# Wavelet Diffusion Posterior Sampling with Frequency Domain Guidance

## Abstract

Inverse imaging problems often involve the reconstruction of high-fidelity signals from noisy and incomplete measurements. Recent advances in diffusion models have achieved promising results for these tasks, yet most approaches operate in the spatial domain and struggle to preserve high-frequency details under noise. To address this issue, we introduce Wavelet diffusion posterior sampling (WDPS), a frequency domain framework that integrates wavelet transforms with posterior sampling. By decomposing images into multiscale frequency subbands, WDPS performs posterior updates adaptively across low- and high-frequency components, enabling more stable sampling trajectories and improved detail recovery. To further enhance robustness, we propose a wavelet-regularized diffusion strategy that dynamically adjusts the influence of frequency-domain constraints during sampling. We demonstrate our approach on both linear and nonlinear inverse problems. We also extend our task to the lensless camera task to show the applicability of our approach. Our results highlight the effectiveness of frequency-domain posterior diffusion as a general and efficient solution to noisy inverse problems.

## 1 Introduction

Inverse problems Kawar et al. (2021; 2022); Chung et al. (2022a;b); Kim et al. (2023); Chung et al. (2023); Daras et al. (2024) are central to computational imaging, where the goal is to reconstruct a clean signal $X^\star$ from noisy and incomplete measurements $Y = \mathcal{A}(X^\star) + \epsilon$, with $\mathcal{A}$ denoting the forward operator and $\epsilon$ representing measurement noise. Such problems arise in a wide range of applications, including medical imaging Webber & Reader (2024), computational photography Nehme & Michaeli (2025), and scientific visualization Yair et al. (2024). However, the inherent ill-posedness of inverse problems Cardoso et al. (2023); Yang et al. (2024), together with the presence of noise, makes high-fidelity recovery particularly challenging.

Recent advances in diffusion models Ho et al. (2020); Song et al. (2020) have demonstrated impressive performance in image generation and inverse problems such as inpainting, super-resolution, and deblurring Chung et al. (2022a;b); Kim et al. (2023); Chung et al. (2023); Daras et al. (2024). A common strategy is *diffusion posterior sampling* (DPS) Chung et al. (2022a), which incorporates measurement consistency constraints during the sampling process. While effective in noiseless or linear cases, spatial-domain posterior sampling often struggles to preserve high-frequency structures under noise Qian et al. (2024); Wan et al. (2023); Song et al. (2025); Li et al. (2025a). This limitation results in blurry reconstructions, loss of fine textures, and unstable convergence when facing complex forward operators. In particular, DPS applies posterior corrections uniformly in the image space, where low- and high-frequency information are entangled. As a result, enforcing data consistency can inadvertently suppress high-frequency structures, leading to a systematic trade-off between fidelity and detail preservation. This motivates the need for a representation that disentangles frequencies and allows more targeted posterior updates.

Images exhibit a natural multi-scale structure: low-frequency components capture global appearance, while high-frequency components encode fine-grained details such as edges and textures. Wavelet transforms Chen et al. (2024); Jin et al. (2025); Huang et al. (2024); Li et al. (2025b) provide an effective decomposition of these components into subbands (LL, LH, HL, HH), enabling frequency-aware processing. Operating in the frequency domain offers the potential to decouple global consistency

from local detail restoration, making it particularly well-suited for inverse problems with strong noise or nonlinear degradations.

In this work, we propose **Wavelet Diffusion Posterior Sampling (WDPS)**, a frequency-domain framework that integrates wavelet transforms with posterior sampling. WDPS decomposes intermediate samples into wavelet subbands, performs posterior updates adaptively across low- and high-frequency components, and reconstructs the image using the inverse wavelet transform. Furthermore, we introduce a wavelet-regularized diffusion strategy, which dynamically adjusts the influence of frequency-domain constraints during sampling, stabilizing the trajectory and improving generalization. In summary, our main contributions can be summarized as follows:

- We introduce WDPS, a framework that performs diffusion posterior sampling directly in the wavelet frequency domain, enabling frequency-aware reconstructions that better preserve fine details under noise.

- We propose a wavelet-regularized sampling scheme that adaptively controls frequency-domain constraints, improving stability and convergence, and further analyze the stability of dynamic wavelet-regularized sampling

- We conduct extensive experiments on FFHQ and ImageNet across diverse tasks (inpainting, super-resolution, etc). We also extend our task to the lensless camera task to show generalization of our approach. WDPS consistently outperforms spatial-domain baselines such as DPS, achieving sharper reconstructions and better quantitative scores (FID, LPIPS, PSNR, and SSIM).

## 2 RELATED WORKS

Diffusion models Ho et al. (2020); Song et al. (2020) define a generative process as the reverse of a noising process, typically described by a variance–preserving SDE

$$dX_t = -\frac{\beta(t)}{2}X_t\,dt + \sqrt{\beta(t)}\,dw, \tag{1}$$

whose reverse SDE includes the data score $\nabla_{X_t} \log p_t(X_t)$. In the Bayesian setting of an inverse problem $Y = \mathcal{A}(X^\star) + \epsilon$, the goal is to sample from the posterior $p(X_0|Y)$, which formally satisfies Bayes' rule

$$p(X_0|Y) = \frac{p(Y|X_0)p(X_0)}{p(Y)}. \tag{2}$$

Naïvely, one might try to modify the reverse SDE to

$$dX_t = \left[-\frac{\beta(t)}{2}X_t - \beta(t)\big(\nabla_{X_t} \log p_t(X_t) + \nabla_{X_t} \log p_t(Y|X_t)\big)\right]dt + \sqrt{\beta(t)}\,dw, \tag{3}$$

so that the drift contains both the prior and the likelihood gradients. However, the likelihood term $\nabla_{X_t} \log p_t(Y|X_t)$ is analytically intractable at intermediate noise levels $t$, so most existing works resort to alternating "unconditional diffusion + projection onto the measurement subspace" under the assumption of negligible noise. This projection can amplify noise and fails for nonlinear operators.

Diffusion Posterior Sampling (DPS) Chung et al. (2022a) circumvents this difficulty by (i) using Tweedie's formula to compute the posterior mean of the clean sample

$$\hat{X}_0 = \frac{1}{\sqrt{\bar{\alpha}(t)}}\Big(X_t + (1 - \bar{\alpha}(t))s_\theta(X_t, t)\Big), \tag{4}$$

which approximates $\mathbb{E}[X_0|X_t]$; and (ii) replacing the intractable expectation in $p(Y|X_t) = \int p(Y|X_0)p(X_0|X_t)dX_0$ with $p(Y|\hat{X}_0)$, leading to a tractable surrogate likelihood. The gradient of this surrogate gives the practical update

$$\nabla_{X_t} \log p_t(X_t|Y) \approx s_\theta(X_t, t) - \rho \nabla_{X_t} \|Y - \mathcal{A}(\hat{X}_0)\|_2^2, \tag{5}$$

or with a weighted norm $\|\cdot\|_\Lambda$ for Poisson noise. Discretizing this yields the DPS algorithms for both Gaussian and Poisson measurements.

While DPS already improves robustness to noise, it still operates entirely in the spatial domain, applying the same posterior correction to all frequencies. In contrast, our proposed Wavelet Diffusion Posterior Sampling (WDPS) decomposes intermediate samples into multi-scale wavelet subbands, applies frequency-adaptive posterior updates and a dynamic wavelet regularizer, thereby addressing the high-frequency suppression and instability observed in spatial-domain posterior sampling. Additional related works are presented in the Appendix.

## 3 METHOD

Our method uses the wavelet transform to transfer the image from the spatial domain to the frequency domain during the diffusion sampling process, and performs the posterior update directly in the frequency domain.

### 3.1 WAVELET-BASED POSTERIOR

We denote by $\mathcal{W} : \mathbb{R}^{H \times W} \rightarrow \mathbb{R}^{H/2 \times W/2 \times 4}$ the discrete wavelet transform (DWT) Heil & Walnut (1989); Sundararajan (2016); Othman & Zeebaree (2020), which decomposes an input image into four sub-bands at each scale. Given an intermediate sample at step $i$, $X'_{i-1} \in \mathbb{R}^{1 \times 3 \times H \times W}$, as shown in Figure 3, the wavelet coefficients are

$$W_{i-1} = \mathcal{W}(X'_{i-1}) = \left( W^{\text{LL}}_{i-1}, W^{\text{LH}}_{i-1}, W^{\text{HL}}_{i-1}, W^{\text{HH}}_{i-1} \right), \tag{6}$$

where

- $W^{\text{LL}}_{i-1}$ (low–low) contains the low-frequency approximation coefficients representing global structure and smooth regions,
- $W^{\text{LH}}_{i-1}$ (low–high) captures vertical high-frequency details,
- $W^{\text{HL}}_{i-1}$ (high–low) captures horizontal high-frequency details, and
- $W^{\text{HH}}_{i-1}$ (high–high) captures diagonal high-frequency details such as edges and fine textures.

Mathematically, for a one-level separable 2-D wavelet transform, we define a *low-pass filter* $h$ and a corresponding *high-pass filter* $g$. For the simplest case of the Haar (Daubechies-1) wavelet, these filters are

$$h = \tfrac{1}{\sqrt{2}}[1,\, 1], \qquad g = \tfrac{1}{\sqrt{2}}[1,\, -1]. \tag{7}$$

Here, $h$ extracts the smooth (low-frequency) components of the signal, while $g$ extracts the detailed (high-frequency) variations. Using these filters, the four sub-bands at each scale can be expressed as

$$W^{\text{LL}} = (X * h * h^{\top}) \downarrow 2, \tag{8}$$

$$W^{\text{LH}} = (X * h * g^{\top}) \downarrow 2, \tag{9}$$

$$W^{\text{HL}} = (X * g * h^{\top}) \downarrow 2, \tag{10}$$

$$W^{\text{HH}} = (X * g * g^{\top}) \downarrow 2, \tag{11}$$

where $X$ is the input image, $*$ denotes convolution along rows/columns, and $\downarrow 2$ indicates downsampling by a factor of 2, i.e., retaining only every second sample along each spatial dimension. After this filtering and downsampling, each sub-band has size $H/2 \times W/2$. Specifically, $W^{\text{LL}}$ contains approximation coefficients capturing global structure, $W^{\text{LH}}$ and $W^{\text{HL}}$ encode vertical and horizontal details, while $W^{\text{HH}}$ captures diagonal details.

During our posterior update we operate directly on $W_{i-1}$ rather than on the raw spatial-domain tensor. Each of the four sub-bands can be processed separately or with sub-band–dependent step sizes, which allows frequency-aware adaptation of the likelihood gradient

$$W'_{i-1} = W_{i-1} - \zeta_i \nabla_{W_i} \|Y - \mathcal{A}(\hat{X}_0)\|_2^2, \tag{12}$$

where $\hat{X}_0$ is the Tweedie estimate of the clean image at the current step. After this frequency-domain posterior correction, we transform the updated coefficients back to the spatial domain using the inverse wavelet transform (IDWT):

$$X_{i-1} = \mathcal{W}^{-1}(W'_{i-1}), \tag{13}$$

where $\mathcal{W}^{-1}(\cdot)$ denotes the **inverse discrete wavelet transform**. Given the four sub-bands $W_{i-1}^{\text{LL}}$, $W_{i-1}^{\text{LH}}$, $W_{i-1}^{\text{HL}}$ and $W_{i-1}^{\text{HH}}$ (each of size $H/2 \times W/2$), the reconstruction can be written as

$$X_{i-1} = \big((W_{i-1}^{\text{LL}}{\uparrow}2) * \tilde{h} * \tilde{h}^{\top}\big) + \big((W_{i-1}^{\text{LH}}{\uparrow}2) * \tilde{h} * \tilde{g}^{\top}\big)$$
$$+ \big((W_{i-1}^{\text{HL}}{\uparrow}2) * \tilde{g} * \tilde{h}^{\top}\big) + \big((W_{i-1}^{\text{HH}}{\uparrow}2) * \tilde{g} * \tilde{g}^{\top}\big), \tag{14}$$

where $\uparrow 2$ denotes upsampling by inserting zeros between samples along each dimension, $*$ denotes convolution, and $\tilde{h}$ and $\tilde{g}$ are the synthesis (inverse) low-pass and high-pass filters corresponding to $h$ and $g$ used in the forward DWT. This reconstruction exactly reverses the DWT, producing a $H \times W$ spatial-domain image from the four $H/2 \times W/2$ sub-bands.

By explicitly separating low- and high-frequency components through $\mathcal{W}(\cdot)$ and $\mathcal{W}^{-1}(\cdot)$, our method preserves significant features and fine details of $X_0$ while suppressing noise and artifacts during sampling. This description matches the detailed procedure summarized in Algorithm 1.

---

**Algorithm 1** Wavelet Diffusion Posterior Sampling (WDPS)

---

**Require:** $N, Y, \{\zeta_i\}_{i=1}^N, \{\tilde{\sigma}_i\}_{i=1}^N$
1: $X_N \sim \mathcal{N}(\mathbf{0}, \mathbf{I})$
2: **for** $i = N - 1$ downto 0 **do**
3: $\quad \hat{s} \leftarrow s_\theta(X_i, i)$
4: $\quad \hat{X}_0 \leftarrow \frac{1}{\sqrt{\bar{\alpha}_i}}\Big(X_i + (1 - \bar{\alpha}_i)\hat{s}\Big)$
5: $\quad z \sim \mathcal{N}(\mathbf{0}, \mathbf{I})$
6: $\quad X'_{i-1} \leftarrow \frac{\sqrt{\alpha_i}(1-\bar{\alpha}_{i-1})}{1-\bar{\alpha}_i}X_i + \frac{\sqrt{\bar{\alpha}_{i-1}}\beta_i}{1-\bar{\alpha}_i}\hat{X}_0 + \tilde{\sigma}_i z$
7: $\quad W_{i-1} \leftarrow \mathcal{W}(X'_{i-1})$
8: $\quad W'_{i-1} \leftarrow W_{i-1} - \zeta_i \nabla_{W_i} \|Y - \mathcal{A}(\hat{X}_0)\|_2^2$
9: $\quad X_{i-1} \leftarrow \mathcal{W}^{-1}(W'_{i-1})$
10: **end for**
11: **return** $\hat{X}_0$

---

### 3.2 WAVELET-REGULARIZED DIFFUSION SAMPLING

Inverse problems are inherently ill-posed, and unconstrained diffusion sampling may accumulate unstable high-frequency artifacts. To stabilize the trajectory, we introduce a *wavelet-based regularization* that adaptively scales the wavelet coefficients at each reverse step using a time-varying strength parameter.

**Wavelet-strength schedule.** For a reverse step index $i \in \{1, \ldots, T\}$, the wavelet strength is defined as

$$r(i; a, b) = \frac{1}{C + \exp\big(i \cdot \frac{a}{b}\big)}, \qquad a > 0, \ b = T, C > 0 \tag{15}$$

where $T$ denotes the total number of diffusion steps, $a$ controls the exponential decay rate, $b$ normalizes the horizon (default $b = 1000$), and $C$ determines the baseline offset of the regularization schedule. For different values of $C$, Figure 4 depicts the regularization schedule under varying baseline offsets. This function produces a smoothly decaying value over the course of sampling: at the first step ($i = 1$), $r \approx 0.125$, while towards the end of sampling $r \to 0$. Thus, the effective regularization is stronger in early steps—where the problem is highly underdetermined—and gradually diminishes as the estimate stabilizes.

**Subband update.** Given the discrete wavelet transform (DWT) of an intermediate iterate $X'_{i-1}$,

$$W_{i-1} = \big(W_{i-1}^{\text{LL}}, W_{i-1}^{\text{LH}}, W_{i-1}^{\text{HL}}, W_{i-1}^{\text{HH}}\big), \tag{16}$$

we preserve the low-frequency band and scale the high-frequency subbands by the wavelet strength:

$$W_{i-1}^{\text{LL}'} = W_{i-1}^{\text{LL}}, \tag{17}$$

$$W_{i-1}^{s'} = r(i; a, b) \cdot W_{i-1}^s, \qquad s \in \{\text{LH}, \text{HL}, \text{HH}\}. \tag{18}$$

The updated coefficients are then transformed back into the spatial domain:

$$X_{i-1} = \mathcal{W}^{-1}(W'_{i-1}). \tag{19}$$

## 3.3 THEORETICAL JUSTIFICATION

The wavelet-strength schedule $r(i)$ is motivated by three key observations. (1) Natural images are approximately sparse in the wavelet domain (Donoho, 2006), where most high-frequency coefficients are negligible or noise-dominated; suppressing them early removes instability without harming structure. (2) The discrete wavelet transform provides a multiresolution analysis (Mallat, 2002), so a decaying $r(i)$ naturally enforces a coarse-to-fine trajectory: global structures first, then fine details. (3) The dynamic schedule stabilizes sampling by imposing strong constraints when the problem is most ill-posed and gradually relaxing them as convergence is reached. This balances stability and detail recovery, justifying wavelet-regularized diffusion sampling. Next, we analyze dynamic regularization stability in the wavelet domain.

**Dynamic regularization stability.** We analyze the reverse-time update in the wavelet domain

$$W'_{i-1} = S_{r_i}\big(W_{i-1} - \zeta_i\,\nabla_W\,\mathcal{L}\big(\mathcal{W}^{-1}(W_{i-1}); y\big)\big), \qquad i = T, \dots, 1, \tag{20}$$

where $\mathcal{L}(x; y) = \|\,\mathcal{A}(x) - y\,\|_2^2$ is the data-fidelity loss, $\mathcal{W}$ is an orthonormal DWT (so $\|\mathcal{W}x\| = \|x\|$), and $S_{r_i}$ is a wavelet-regularization operator parameterized by a decaying schedule $r_i = r(i; a, b)$.

**Theorem 1 (Stability of dynamic wavelet-regularized sampling)** *Assume:*

1. *(**Smooth forward model**) $\mathcal{A} : \mathbb{R}^{H \times W \times 3} \to \mathbb{R}^m$ is $L_A$-Lipschitz and has $L_\nabla$-Lipschitz Jacobian.*

2. *(**Wavelet isometry**) $\mathcal{W}$ is orthonormal, hence $\|\mathcal{W}x\|_2 = \|x\|_2$ and $\|\nabla_W\mathcal{L}(\mathcal{W}^{-1}(\cdot); y)\| \leq L\,\|\cdot\|$ for some $L > 0$.*

3. *(**Nonexpansive regularizer**) For every $r \in [0, r_1]$, the operator $S_r : \mathbb{R}^{H/2 \times W/2 \times 4} \to \mathbb{R}^{H/2 \times W/2 \times 4}$ is nonexpansive: $\|S_r(u) - S_r(v)\|_2 \leq \|u - v\|_2$. This holds for (i) High-Frequency gating $S_r(u) = (u^{LL}, r\,u^{LH}, r\,u^{HL}, r\,u^{HH})$ with $r \in [0, 1]$; and (ii) soft-thresholding $S_r(u) = \mathrm{soft}(u; \tau_r)$, which is firmly nonexpansive as a proximal map.*

4. *(**Step size**) $0 < \zeta_i \leq 2/L$ for all $i$ (e.g., $\zeta_i \leq 1/L$ is sufficient), where $L$ is a Lipschitz constant of $\nabla_W\mathcal{L}(\mathcal{W}^{-1}(\cdot); y)$.*

*Then each one-step map*

$$\mathcal{T}_i(\cdot; y) = S_{r_i}\big(\mathrm{Id} - \zeta_i\nabla_W\mathcal{L}(\mathcal{W}^{-1}(\cdot); y)\big) \tag{21}$$

*is nonexpansive in the iterate and Lipschitz in the measurement:*

$$\big\|\mathcal{T}_i(U; y) - \mathcal{T}_i(V; y)\big\|_2 \leq \|U - V\|_2, \qquad \big\|\mathcal{T}_i(U; y) - \mathcal{T}_i(U; y')\big\|_2 \leq c_i\,\|y - y'\|_2, \tag{22}$$

*for some constants $c_i = \zeta_i\,\mathrm{Lip}_y(\nabla_W\mathcal{L})$. Consequently, for trajectories driven by the same noise $X_T$ and two measurements $y, y'$,*

$$\|X_0(y) - X_0(y')\|_2 \leq \Big(\sum_{i=1}^{T} c_i\Big)\,\|y - y'\|_2, \tag{23}$$

*i.e., the final reconstruction is* Lipschitz-stable *with respect to perturbations in $y$. If, in addition, each gradient step is* strictly *contractive (e.g., by strong convexity in a local basin or smaller $\zeta_i$), the bound tightens to a decaying product form.*

Theorem 1 shows that the update rule is nonexpansive in the iterate and Lipschitz-stable with respect to the measurement, i.e., small perturbations in $y$ do not cause large deviations in the reconstruction. The dynamic wavelet strength $r(i)$ is central: large values in early steps suppress unstable high-frequency components under the ill-posed forward operator, while its decay later permits fine details to emerge without sacrificing stability. This justifies the coarse-to-fine behavior of our method—stable against noise initially, yet flexible enough to refine textures and edges as sampling progresses.

Table 1: Quantitative results (FID ↓, LPIPS ↓) on FFHQ Dataset across different tasks.

| Methods | Super Resolution | | Inpainting | | Gaussian Blur | | Motion Blur | |
|---|---|---|---|---|---|---|---|---|
| | FID ↓ | LPIPS ↓ | FID ↓ | LPIPS ↓ | FID ↓ | LPIPS ↓ | FID ↓ | LPIPS ↓ |
| **WDPS (ours)** | **32.74** | 0.1979 | **27.11** | **0.1106** | **26.12** | 0.1501 | **28.17** | **0.1804** |
| DPS Chung et al. (2022a) | 36.50 | **0.1932** | 34.16 | 0.1115 | 29.78 | **0.1461** | 31.68 | 0.1873 |
| DDRM Kawar et al. (2022) | 62.15 | 0.294 | 69.71 | 0.587 | 74.92 | 0.332 | – | – |
| MCG Chung et al. (2022b) | 87.64 | 0.520 | 29.26 | 0.286 | 101.2 | 0.340 | 310.5 | 0.702 |
| PnP-ADMM Chan et al. (2016) | 66.52 | 0.353 | 123.6 | 0.692 | 90.42 | 0.441 | 89.08 | 0.405 |
| Score-SDE Song et al. (2020) | 96.72 | 0.563 | 76.54 | 0.612 | 109.0 | 0.403 | 292.2 | 0.657 |
| ADMM-TV Wahlberg et al. (2012) | 110.6 | 0.428 | 181.5 | 0.463 | 186.7 | 0.507 | 152.3 | 0.508 |

Table 2: Quantitative results (FID ↓, LPIPS ↓) on ImageNet dataset across different tasks.

| Methods | Motion Blur | | Gaussian Blur | | Inpainting | | Super Resolution | |
|---|---|---|---|---|---|---|---|---|
| | FID ↓ | LPIPS ↓ | FID ↓ | LPIPS ↓ | FID ↓ | LPIPS ↓ | FID ↓ | LPIPS ↓ |
| **WDPS (ours)** | **42.80** | **0.2201** | **39.49** | **0.2548** | **39.62** | **0.1953** | **50.55** | **0.3319** |
| DPS Chung et al. (2022a) | 44.67 | 0.2450 | 44.93 | 0.2747 | 49.35 | 0.2073 | 54.10 | 0.3319 |
| DDRM Kawar et al. (2022) | – | – | 63.02 | 0.427 | 45.95 | 0.245 | 59.57 | 0.339 |
| MCG Chung et al. (2022b) | 186.9 | 0.758 | 95.04 | 0.550 | 39.74 | 0.330 | 144.5 | 0.637 |
| PnP-ADMM Chan et al. (2016) | 89.76 | 0.483 | 100.6 | 0.519 | 78.02 | 0.367 | 97.22 | 0.433 |
| Score-SDE Song et al. (2020) | 98.25 | 0.591 | 123.0 | 0.667 | 54.07 | 0.315 | 170.7 | 0.701 |
| ADMM-TV Wahlberg et al. (2012) | 138.8 | 0.525 | 155.7 | 0.588 | 87.69 | 0.319 | 130.9 | 0.523 |

# 4 EXPERIMENT

## 4.1 EXPERIMENTAL SETUP

We evaluate our Wavelet Diffusion Posterior Sampling (WDPS) framework on two widely used datasets that exhibit diverging characteristics: FFHQ (256×256) (Karras et al., 2019) and ImageNet (256×256) (Deng et al., 2009). For ImageNet, we adopt the pre-trained diffusion model released by Dhariwal & Nichol (2021) and directly use it without task-specific finetuning. For FFHQ, we sample with the pretrained model used in Chung et al. (2022a).

Forward measurement operators are specified as follows: (i) **Inpainting:** We use a 128×128 box mask following Chung et al. (2022a), and in random-type inpainting we mask out 92% of the pixels across all RGB channels. (ii) **Super-resolution:** Bicubic downsampling is applied. (iii) **Gaussian blur:** Convolution with a Gaussian kernel of size 61×61 and standard deviation 3.0. (iv) **Motion blur:** Convolution with randomly generated motion kernels of size 61×61 and intensity 0.5. (v) **Nonlinear deblurring:** We employ a neural network-based forward model following Chung et al. (2022a). (vi) **Lensless camera:** Following Antipa et al. (2017); Monakhova et al. (2019); Hung et al. (2025), we apply a diffuser point spread function (PSF) as the forward operator to simulate lensless measurements. For noise models, Gaussian noise with $\sigma = 0.05$ is added to the measurement domain, and the Poisson noise level is set to $\lambda = 1.0$.

To assess reconstruction quality, we employ both pixel-level and perceptual metrics. Pixel-level fidelity is measured using Peak Signal-to-Noise Ratio (PSNR, ↑) and Structural Similarity Index Measure (SSIM, ↑). Perceptual quality is evaluated with Fréchet Inception Distance (FID, ↓) and Learned Perceptual Image Patch Similarity (LPIPS, ↓). These complementary metrics ensure a comprehensive comparison between WDPS and the baseline Diffusion Posterior Sampling (DPS).

## 4.2 RESULTS ON LINEAR INVERSE PROBLEM

Table 3: Quantitative results (PSNR ↑, SSIM ↑) on ImageNet dataset across different tasks.

| Methods | Motion Blur | | Gaussian Blur | | Inpainting | | Super Resolution | |
|---|---|---|---|---|---|---|---|---|
| | PSNR ↑ | SSIM ↑ | PSNR ↑ | SSIM ↑ | PSNR ↑ | SSIM ↑ | PSNR ↑ | SSIM ↑ |
| **WDPS (ours)** | **27.68** | **0.7654** | **23.96** | **0.6424** | 27.98 | 0.7762 | **22.63** | **0.6114** |
| DPS Chung et al. (2022a) | 24.63 | 0.6735 | 23.30 | 0.6126 | **28.09** | **0.7763** | 22.55 | 0.6019 |

Table 4: Quantitative results (PSNR ↑, SSIM ↑) on FFHQ dataset across different tasks.

| Methods | Motion Blur | | Gaussian Blur | | Inpainting | | Super Resolution | | Nonlinear Blur | |
|---|---|---|---|---|---|---|---|---|---|---|
| | PSNR ↑ | SSIM ↑ | PSNR ↑ | SSIM ↑ | PSNR ↑ | SSIM ↑ | PSNR ↑ | SSIM ↑ | PSNR ↑ | SSIM ↑ |
| **WDPS (ours)** | **23.34** | **0.6551** | **25.00** | 0.6894 | **29.48** | 0.8305 | **23.61** | 0.6615 | **23.15** | **0.6466** |
| DPS Chung et al. (2022a) | 22.54 | 0.6311 | 24.89 | **0.6884** | 29.39 | **0.8347** | 23.59 | **0.6622** | 22.82 | 0.6322 |

We evaluate the proposed WDPS method against the DPS baseline as well as several representative baselines, including DDRM Kawar et al. (2022), MCG Chung et al. (2022b), PnP-ADMM Chan et al. (2016), Score-SDE Song et al. (2020), ADMM-TV Wahlberg et al. (2012), BKS-based approaches Tran et al. (2021), and FPS Dou & Song (2024). The quantitative results across five common image restoration tasks on the FFHQ dataset are reported in Table 1, using Fréchet Inception Distance (FID, ↓) and Learned Perceptual Image Patch Similarity (LPIPS, ↓). WDPS achieves consistently lower FID than DPS across all tasks, with particularly large margins under motion blur (28.17 vs. 31.68), Gaussian blur (26.12 vs. 29.78), and inpainting (27.11 vs. 34.16). Perceptual quality, measured by LPIPS, is also generally improved (e.g., 0.1804 vs. 0.1873 for motion blur; 0.1106 vs. 0.1115 for inpainting). In super-resolution, WDPS shows a clear FID improvement (32.74 vs. 36.50), while DPS attains a slightly better LPIPS.

We further evaluate WDPS on the ImageNet validation set with 1,000 images, using pre-trained diffusion models without task-specific fine-tuning. The results across five restoration tasks are reported in Table 2 and Table 5. WDPS consistently outperforms DPS across all metrics. In motion blur, it improves PSNR by over 3 dB (27.68 vs. 24.63) while also yielding better SSIM and LPIPS. Similar improvements are observed for Gaussian blur, where FID decreases from 44.93 to 39.49.

Inpainting shows a large perceptual gain (FID 39.62 vs. 49.35), despite DPS achieving a marginally higher PSNR. Super-resolution further confirms the advantage, with WDPS reducing FID (50.55 vs. 54.10) while maintaining competitive LPIPS.

Table 5: Quantitative results on ImageNet Nonlinear Blur task.

| Methods | FID ↓ | LPIPS ↓ |
|---|---|---|
| **WDPS (ours)** | **65.68** | **0.3679** |
| FPS Dou & Song (2024) | 196.07 | 0.7423 |
| DPS Chung et al. (2022a) | 78.54 | 0.4190 |
| MGPS Moufad et al. (2024a) | 110 | 0.43 |

Overall, WDPS demonstrates a consistent advantage over DPS and outperforms traditional baselines by a wide margin, delivering reconstructions that are perceptually closer to the ground truth. WDPS generalizes well beyond FFHQ, consistently surpassing DPS on ImageNet, especially in perceptual metrics (FID and LPIPS), and scaling effectively to diverse and challenging restoration tasks.

### 4.3 RESULTS ON NONLINEAR INVERSE PROBLEM

For nonlinear blur, WDPS achieves the best quantitative results, improving both FID (65.68 vs. 78.54) and LPIPS (0.368 vs. 0.419), demonstrating robustness to complex degradations. In addition, WDPS surpasses both DPS (35.11 vs. 38.50) and FPS (196.1 FID), highlighting its robustness to complex degradations.

Overall, WDPS generalizes well beyond FFHQ, consistently surpassing DPS on ImageNet, especially in perceptual metrics (FID and LPIPS), and scaling effectively to diverse and challenging restoration tasks. Below, we provide potential explanations why our method excels in non-linear inverse problems. *(i) Subband-adaptive posterior corrections.* WDPS computes the posterior update in the wavelet domain, decomposing $x$ into low- and high-frequency subbands and applying band-specific step sizes. This subband conditioning decouples the ill-scaled directions induced by $\nabla A(x)$, preventing the likelihood gradient from disproportionately damping

Table 6: Quantitative results on FFHQ Nonlinear Blur task.

| Methods | FID ↓ | LPIPS ↓ |
|---|---|---|
| **WDPS (ours)** | **35.11** | **0.2203** |
| DPS Chung et al. (2022a) | 38.50 | 0.2291 |
| FPS Dou & Song (2024) | 196.5 | 0.701 |
| BKS-styleGAN2 Tran et al. (2021) | 63.18 | 0.407 |
| BKS-generic Tran et al. (2021) | 141.0 | 0.640 |
| MCG Chung et al. (2022b) | 180.1 | 0.695 |
| MGPS Moufad et al. (2024a) | 50.8 | 0.23 |

high-frequency coefficients. Edges and fine textures receive well-calibrated corrections while low-frequency structure remains stable, yielding better measurement fits.

*(ii) Coarse-to-fine scheduling via dynamic wavelet regularization.* Early diffusion steps are the most ill-posed: noise levels are high, the posterior is broad, and nonlinearities in $A(\cdot)$ can amplify spurious

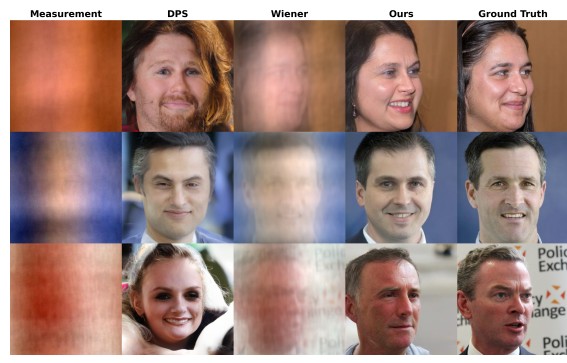

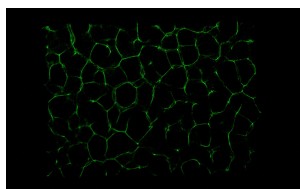

(a) Reconstructions comparing DPS and WDPS with ground-truth.      (b) Visualization of PSF.

Figure 1: Evaluation on the DiffuserCam dataset. (a) Example reconstructions with DPS and WDPS compared against ground truth. (b) Corresponding point spread function (PSF). The image is enhanced for better visualization.

Table 7: Quantitative results (FID ↓, PSNR ↑, SSIM ↑, LPIPS ↓) on DiffuserCam Task.

| Methods | FID ↓ | PSNR ↑ | SSIM ↑ | LPIPS ↓ |
|---|---|---|---|---|
| **WDPS (ours)** | **36.77** | **16.30** | **0.4632** | **0.3693** |
| DPS Chung et al. (2022a) | 47.24 | 14.29 | 0.3842 | 0.4220 |
| FPS-SMC Dou & Song (2024) | 330.77 | 8.88 | 0.1915 | 0.8518 |
| ADMM Wahlberg et al. (2012) | 272.32 | 14.42 | 0.4572 | 0.6499 |
| Wiener Deconvolution | 261.10 | 15.38 | 0.5357 | 0.6451 |

detail. WDPS introduces a time-varying wavelet regularizer that is strong at the beginning (suppressing unstable high-frequency digressions) and is gradually relaxed to let fine-scale information emerge as the noise level decays. This schedule acts as an implicit continuation method—first solving an easier, smoothed problem and then homotopying to the fully detailed reconstruction—thereby reducing artifacts that commonly plague nonlinear inverse solvers.

*(iii) Improved gradient conditioning and stability.* Performing the data-consistency correction on $\hat{x}_0$ in the wavelet domain improves the conditioning of the update map. The resulting reverse dynamics are less sensitive to local Lipschitz spikes of $A(\cdot)$ and to moderate operator/model mismatch. Empirically, this manifests as lower variance across runs, fewer catastrophic samples, and improved sample efficiency (fewer steps required to reach a given quality).

*(iv) Robustness to model mismatch.* Realistic nonlinear degradations rarely match the training-time assumptions of baseline samplers. Because WDPS regularizes *frequencies* rather than *pixels*, it tolerates moderate mismatches in the forward map by preserving the spectral statistics of natural images even when the likelihood term is slightly mis-specified. Spatial-only corrections, in contrast, tend to either oversmooth to satisfy data consistency or overfit noise amplified through $A(\cdot)$.

### 4.4 RESULTS ON LENSELESS IMAGE

We evaluate the proposed WDPS method against the DPS baseline on lensless imaging, where measurements are formed by convolving the original image with a point spread function (PSF). The PSF acts as a perturbation kernel that encodes the optical response of the lensless system, making the inverse problem highly ill-posed. For this task, we adopt the PSF provided by Hung et al. (2025) and apply it to FFHQ images using the same pre-trained model as in previous experiments. Quantitative results are reported in Table 7 while qualitative reconstructions are shown in Figure 1. Under this challenging forward operator, DPS produces severely distorted and unrealistic facial reconstructions. In contrast, WDPS effectively suppresses these artifacts and yields reconstructions that closely resemble the ground-truth images, demonstrating its robustness in lensless imaging scenarios.

The results confirm that the WDPS method outperforms the baseline methods in both quantitative metrics and perceptual quality. The ability to leverage frequency-domain information through wavelet transforms and posterior sampling enhances the reconstruction quality, especially for tasks requiring

fine detail preservation such as nonlinear deblurring. Our approach sets a new benchmark for solving inverse problems and opens avenues for future work in frequency-domain diffusion models.

### 4.5 ABLATION STUDY

We conduct an ablation study on the proposed dynamic wavelet regularization schedule. Results are provided in Tables 8.

On ImageNet, the dynamic schedule consistently improves performance across all metrics, confirming its effectiveness in stabilizing training and preserving high-frequency details in diverse and challenging settings. On FFHQ, the gains are generally more modest for tasks such as blur or inpainting, since the dataset consists of high-quality, homogeneous face images where the diffusion prior alone already provides strong reconstructions. An exception is super-resolution, where the task is inherently ill-posed and requires recovery of high-frequency details; here the schedule plays a crucial role, leading to dramatic improvements (e.g., FID 32.74 vs. 192.41). For physics-based imaging (DiffuserCam), the benefits are smaller and task-dependent: WDPS improves FID and PSNR,

Table 8: Quantitative results (FID, LPIPS) for the first three tasks.

| Task | Method | FID ↓ | LPIPS ↓ |
|---|---|---|---|
| Motion Blur | strength | **48.2538** | **0.3354** |
| | no-strength | 187.3271 | 0.7754 |
| Gaussian Blur | strength | **39.4935** | **0.2548** |
| | no-strength | 182.6592 | 0.7586 |
| Inpainting | strength | **39.6198** | **0.1953** |
| | no-strength | 128.5177 | 0.6185 |

while SSIM and LPIPS may not always benefit. Due to page limits, additional ablation results are provided in Appendix E.

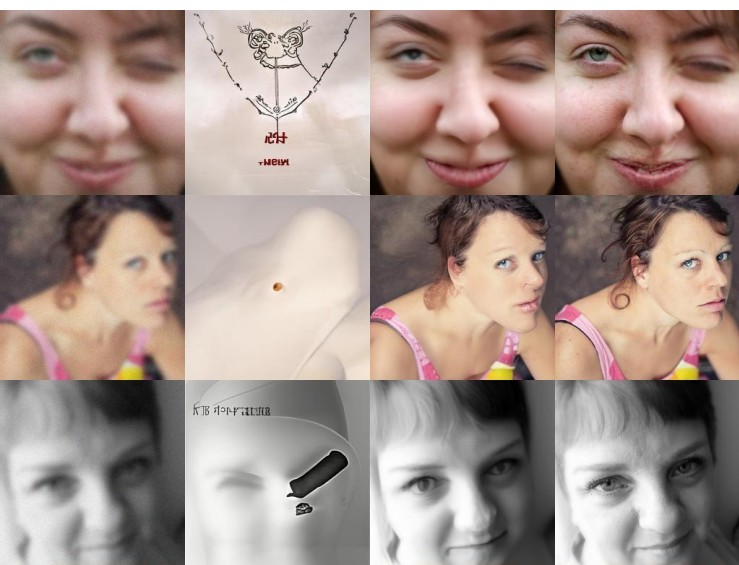

Figure 2: Gaussian Blur ablation on ImageNet—rows: **Measurement**, **No-strength**, **Strength**, **Ground Truth**.

## 5 CONCLUSION

We present a frequency-domain-based posterior sampling approach that leverages frequency features to improve performance on image inverse problems. By operating in the frequency domain, this method enables more effective reconstruction and opens new avenues for future research into frequency-aware image processing techniques. Our approach, Wavelet-based Diffusion Posterior Sampling (WDPS), demonstrates strong performance compared to state-of-the-art methods on the evaluated tasks, highlighting its potential for broad applicability across various domains.

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

# A  ADDITIONAL RELATED WORKS

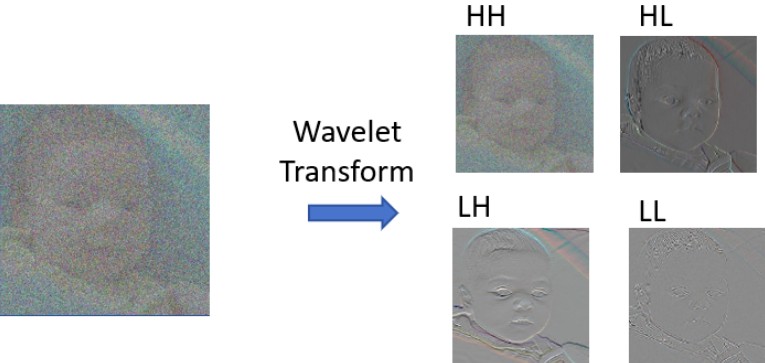

Figure 3: Wavelet Transform.

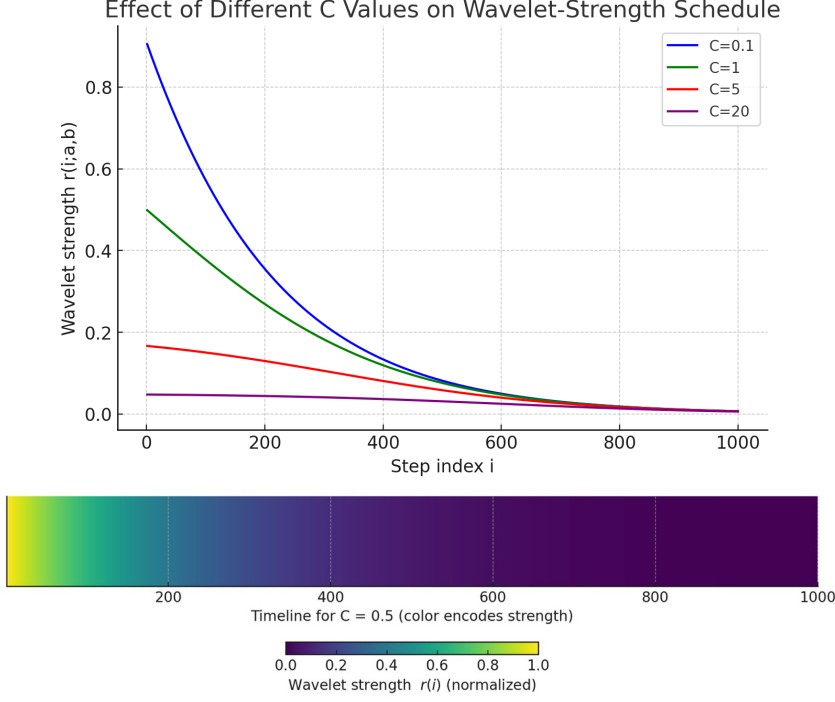

Figure 4: Regularization Schedule Under Different Baseline Offset.

## A.1  DIFFUSION MODEL ON FREQUENCY DOMAIN

Recent research Qian et al. (2024); Wan et al. (2023); Song et al. (2025); Li et al. (2025a); Phung-Ngoc et al. (2025); De Paepe et al. (2025b;a) has highlighted the advantages of applying diffusion models in the frequency domain. Such frequency-based diffusion models typically achieve superior generative performance compared to traditional methods operating solely in the spatial (image) domain. For instance, Qian et al. (2024) reinterpret iterative denoising as an optimization process and introduce Moving Average Sampling in the Frequency domain (MASF): instead of simply averaging

intermediate denoised samples, they first map them back to data space and then perform a moving average separately on different frequency components. This frequency-aware ensembling stabilizes the sampling trajectory and improves both unconditional and conditional diffusion models with negligible extra cost. Similarly, Wan et al. (2023) present the first text-conditioned human motion generation method in the frequency domain of motions, encoding motion sequences into a compact phase space that preserves high-frequency periodic details and then using a conditional diffusion model to predict these parameters from text, enabling smooth transitions and diverse long-term motion synthesis. Song et al. (2025) tackle underwater image enhancement by proposing a two-stage frequency-domain latent diffusion model (FD-LDM): a lightweight parameter estimation network first corrects color bias, and then high- and low-frequency priors are extracted and fused with a refined latent diffusion model to further enhance degraded underwater images. Li et al. (2025a) address change detection in remote sensing by proposing DSFI-CD, which uses a conditional denoising diffusion model to generate pseudo-images and introduces a spatial–frequency interaction module plus an edge-enhanced module to better capture high-frequency edge information and improve robustness.

Utilizing frequency-domain methods naturally aligns with a divide-and-conquer approach, enabling inverse problems to be addressed independently within distinct frequency bands—such as high-frequency and low-frequency components. However, existing frequency-based approaches often emphasize improvements in model architectures rather than exploring novel sampling techniques.

### A.2 WAVELET TRANSFORMS

Wavelet transforms have been widely incorporated into generative modeling frameworks Chen et al. (2024); Jin et al. (2025); Huang et al. (2024); Li et al. (2025b). For example, Chen et al. (2024) use a discrete wavelet transform with a conditional diffusion model to generate accurate multi-modal pedestrian trajectories. Similarly, Jin et al. (2025) proposes MWT-Diff for low-light image enhancement, replacing convolutional up/down-sampling with multi-layer wavelet transforms in a U-Net to extract high-order multi-scale features and fuse them for reconstruction. These works show how wavelet–frequency representations can enhance diffusion models; however, our WDPS likewise exploits multi-scale information but applies it to posterior updates for inverse imaging tasks.

### A.3 LENSLESS CAMERA TASK

Mask-based lensless imaging systems Antipa et al. (2017); Monakhova et al. (2019); Pan et al. (2022); Boominathan et al. (2020); Hung et al. (2025) provide an appealing alternative to traditional lensed cameras due to their compact design, reduced weight, and mechanical simplicity. Instead of directly forming an image through optical lenses, these systems capture multiplexed light patterns on a sensor, which must then be computationally inverted to recover the scene. This inversion is inherently an ill-posed problem, making lensless imaging a natural testbed for evaluating reconstruction algorithms.

In our experiments, we do not build a new physical system; instead, we adopt the forward operator defined by a lensless camera point spread function (PSF) and apply it to standard datasets such as FFHQ. This allows us to simulate realistic lensless measurements while retaining controlled ground-truth references. Compared to conventional model-based reconstructions, which often require heavy computation, precise calibration, and hand-crafted denoisers, our frequency-domain diffusion framework provides a more generalizable and robust solution under these challenging lensless conditions.

## B  DETAILED PROOF OF THEOREM 1.

We denote the loss by
$$\mathcal{L}(x; y) = \|\mathcal{A}(x) - y\|_2^2, \qquad x = \mathcal{W}^{-1}(W), \tag{24}$$
and the one-step operator as
$$\mathcal{T}_i(\cdot; y) \;=\; S_{r_i}\big(\,\mathrm{Id} - \zeta_i \nabla_W \,\mathcal{L}(\mathcal{W}^{-1}(\cdot); y)\big). \tag{25}$$
Here $\mathcal{W}$ is an orthonormal discrete wavelet transform, and $S_{r_i}$ denotes the dynamic wavelet regularization operator (either High-Frequency gating or soft-thresholding). Our goal is to show: 1) $\mathcal{T}_i$ is

nonexpansive in its input; 2) $\mathcal{T}_i$ is Lipschitz with respect to the measurement $y$; 3) the composition of $T$ steps yields global Lipschitz stability of the reconstruction $X_0(y)$ with respect to $y$.

**Step 1. Wavelet isometry.** Since $\mathcal{W}$ is orthonormal, we have

$$\|W_1 - W_2\|_2 = \|\mathcal{W}^{-1}(W_1) - \mathcal{W}^{-1}(W_2)\|_2, \tag{26}$$

and by the chain rule

$$\nabla_W \mathcal{L}(\mathcal{W}^{-1}(W); y) = \mathcal{W} \nabla_x \mathcal{L}(x; y), \quad x = \mathcal{W}^{-1}(W). \tag{27}$$

Thus the Lipschitz constant of $\nabla_W \mathcal{L}$ equals that of $\nabla_x \mathcal{L}$.

**Step 2. Lipschitz constant of the gradient.** If $\mathcal{A}$ is linear, $\mathcal{A}(x) = Ax$, then

$$\mathcal{L}(x; y) = \|Ax - y\|_2^2, \qquad \nabla_x \mathcal{L}(x; y) = 2A^\top (Ax - y). \tag{28}$$

Therefore $\nabla_x \mathcal{L}$ is $L$-Lipschitz with $L = 2\|A\|_2^2$. The same constant applies in the wavelet domain: $\nabla_W \mathcal{L}$ is $L$-Lipschitz.

**Step 3. Nonexpansiveness of the gradient step.** Define the gradient descent step

$$G_i(W; y) = W - \zeta_i \nabla_W \mathcal{L}(\mathcal{W}^{-1}(W); y). \tag{29}$$

By standard results (Baillon–Haddad or cocoercivity of smooth convex functions), if $0 < \zeta_i \leq 2/L$, then

$$\|G_i(U; y) - G_i(V; y)\|_2 \leq \|U - V\|_2. \tag{30}$$

Hence $G_i(\cdot; y)$ is nonexpansive.

**Step 4. Nonexpansiveness of the regularizer.** By assumption, $S_{r_i}$ is nonexpansive:

$$\|S_{r_i}(U) - S_{r_i}(V)\|_2 \leq \|U - V\|_2. \tag{31}$$

This is satisfied both by (i) *High-Frequency gating* $S_{r_i}(u) = (u^{\text{LL}}, r_i u^{\text{LH}}, r_i u^{\text{HL}}, r_i u^{\text{HH}})$ with $r_i \in [0, 1]$; (ii) *Soft-thresholding*, which is firmly nonexpansive as a proximal map.

**Step 5. One-step nonexpansiveness and Lipschitz continuity in $y$.** Since $\mathcal{T}_i = S_{r_i} \circ G_i$, both operators being nonexpansive, their composition is nonexpansive:

$$\|\mathcal{T}_i(U; y) - \mathcal{T}_i(V; y)\|_2 \leq \|U - V\|_2. \tag{32}$$

For Lipschitz continuity in the measurement, consider

$$\|\mathcal{T}_i(U; y) - \mathcal{T}_i(U; y')\|_2 \leq \zeta_i \|\nabla_W \mathcal{L}(\mathcal{W}^{-1}(U); y) - \nabla_W \mathcal{L}(\mathcal{W}^{-1}(U); y')\|_2. \tag{33}$$

By the mean-value inequality, the RHS is bounded by $c_i \|y - y'\|_2$ for some constant $c_i = \zeta_i \operatorname{Lip}_y(\nabla_W \mathcal{L})$.

**Step 6. Stability over $T$ steps.** Let $X_T$ be the common initialization (noise) and $X_0(y)$, $X_0(y')$ the final reconstructions under two measurements. Applying the nonexpansiveness and Lipschitz property iteratively, we obtain

$$\|X_0(y) - X_0(y')\|_2 \leq \Big( \sum_{i=1}^{T} c_i \Big) \|y - y'\|_2. \tag{34}$$

Thus the reconstruction is Lipschitz-stable with respect to measurement perturbations.

**Step 7. Strict contraction case.** If each gradient step is strictly contractive (e.g., due to strong convexity in a local basin or by taking smaller $\zeta_i$), the bound improves to a product form via a discrete Grönwall inequality, yielding exponentially decaying error propagation across steps.

$\square$

## C  STATEMENT ON THE USE OF LARGE LANGUAGE MODELS

We used a large language model strictly for editorial polishing of writing (e.g., grammar, concision, and clarity). The tool was not used to generate substantive content including research ideas and data analyses. No sensitive or identifying information was shared. All LLM-suggested edits were screened by the authors for accuracy and appropriateness, and the final text reflects authors' intent and judgment. The authors accept full responsibility for the integrity and originality of the manuscript.

## D    RUNTIME COMPARISON

To further evaluate the efficiency of our method, we report the average runtime for sampling a single image. All experiments are conducted on an NVIDIA A6000 GPU. As shown in Table 10, WDPS runs slightly slower than DPS on both datasets. This additional cost comes from the wavelet-domain regularization, but it is marginal compared to the overall runtime and is justified by the improvement in reconstruction quality.

## E    ABLATION STUDY ON REGULARIZATION STRENGTH

We evaluate the proposed dynamic wavelet regularization schedule (reported in the first row of each block) against a baseline without regularization (second row), with results presented in Tables 11, 13, and 12. On ImageNet, the dynamic schedule consistently enhances performance across all metrics, demonstrating its ability to stabilize training and preserve high-frequency details in complex and varied scenarios. For FFHQ, improvements are generally more modest in tasks such as deblurring and inpainting, since the dataset contains high-quality, homogeneous facial images where the diffusion prior alone already yields strong reconstructions. A notable exception is super-resolution, where recovering high-frequency structure is essential; in this setting the schedule is critical, producing substantial gains (e.g., FID 32.74 vs. 192.41). For physics-based imaging (DiffuserCam), the impact is more task-specific: WDPS boosts FID and PSNR, whereas SSIM and LPIPS do not always improve. Taken together, these findings suggest that dynamic wavelet regularization provides broad benefits for ill-posed inverse problems, delivering particularly large improvements in high-frequency recovery tasks, while its advantages in physics-driven settings are more nuanced.

Table 11: Quantitative Results on FFHQ Dataset Across Different Tasks

| Task | Method | FID ↓ | PSNR ↑ | SSIM ↑ | LPIPS ↓ |
|------|--------|-------|--------|--------|---------|
| Motion Blur | strength | 28.17 | **23.34** | **0.6551** | **0.1804** |
| | no-strength | **27.6417** | 22.8215 | 0.6434 | 0.1843 |
| Gaussian Blur | strength | 26.12 | 25.00 | 0.6894 | 0.1513 |
| | no-strength | **26.0683** | **25.0087** | **0.6903** | **0.1508** |
| Inpainting | strength | **27.11** | **29.39** | **0.8305** | **0.1106** |
| | no-strength | 27.6930 | 28.6620 | 0.8201 | 0.1219 |
| Super Resolution | strength | **32.74** | **23.61** | **0.6615** | **0.1979** |
| | no-strength | 192.4145 | 6.8106 | 0.3532 | 0.6352 |
| Nonlinear Blur | strength | **35.11** | **23.15** | **0.6466** | 0.2203 |
| | no-strength | 38.7448 | 23.0002 | 0.6445 | **0.2195** |

Table 12: Quantitative Results on DiffuserCam Task(ablation)

| Task | Method | FID ↓ | PSNR ↑ | SSIM ↑ | LPIPS ↓ |
|------|--------|-------|--------|--------|---------|
| DiffuserCam | strength | **36.77** | **16.30** | 0.4632 | 0.3693 |
| | no-strength | 40.1778 | 16.2401 | **0.4648** | **0.3690** |

Table 9: Quantitative results (PSNR ↑, SSIM ↑) on ImageNet Nonlinear Blur task.

| Methods | PSNR ↑ | SSIM ↑ |
|---------|--------|--------|
| **WDPS (ours)** | 21.94 | **0.5818** |
| FPS Dou & Song (2024) | 12.66 | 0.2947 |
| DPS Chung et al. (2022a) | 21.72 | 0.5512 |
| MGPS Moufad et al. (2024b) | **22.4** | 0.57 |

Table 10: Average runtime for sampling a single image (minutes:seconds) on an NVIDIA A6000 GPU.

| Method | FFHQ | ImageNet |
|--------|------|----------|
| DPS | 1:02 | 3:19 |
| WDPS | 1:12 | 3:25 |

| Method | PSNR ↑ | SSIM ↑ | FID ↓ | LPIPS ↓ |
|--------|--------|--------|-------|---------|
| DAPS (CVPR 2025) | 27.64 | 0.721 | 82.54 | 0.198 |
| WDPS (ours) | 25.00 | 0.689 | **26.12** | **0.151** |

Table 14: Comparison with DAPS on FFHQ Gaussian Blur. DAPS yields higher pixel-level fidelity (PSNR/SSIM), while WDPS achieves substantially better perceptual quality (FID/LPIPS).

Table 13: Quantitative Results on ImageNet Dataset Across Different Tasks

| Task | Method | FID ↓ | PSNR ↑ | SSIM ↑ | LPIPS ↓ |
|------|--------|-------|--------|--------|---------|
| Motion Blur | strength | **48.2538** | **21.3515** | **0.5706** | **0.3354** |
| | no-strength | 187.3271 | 8.5685 | 0.3174 | 0.7754 |
| Gaussian Blur | strength | **39.4935** | **23.9636** | **0.6424** | **0.2548** |
| | no-strength | 182.6592 | 8.9774 | 0.3325 | 0.7586 |
| Inpainting | strength | **39.6198** | **27.9836** | **0.7762** | **0.1953** |
| | no-strength | 128.5177 | 13.2764 | 0.4284 | 0.6185 |
| Super Resolution | strength | **50.5472** | **22.6251** | **0.6114** | 0.7569 |
| | no-strength | 191.6817 | 5.4544 | 0.2952 | **0.8446** |
| Nonlinear Blur | strength | **65.6763** | **21.9391** | **0.5818** | **0.3679** |
| | no-strength | 187.5502 | 9.0486 | 0.3239 | 0.7541 |

## F    COMPARISON WITH DAPS

We additionally report results from DAPS Zhang et al. (2025) on FFHQ Gaussian Blur in Table 14. DAPS achieves stronger pixel-level fidelity (PSNR 27.64, SSIM 0.721) than WDPS (PSNR 25.00, SSIM 0.689), which suggests that DAPS preserves low-frequency content more closely under this setting. However, WDPS substantially outperforms DAPS on perceptual metrics, with much lower FID (26.12 vs. 82.54) and LPIPS (0.151 vs. 0.198). This contrast appears to indicate that, while DAPS favors pixel-aligned reconstructions, WDPS better captures perceptual realism and high-frequency details, aligning with our goal of frequency-aware posterior sampling in challenging restoration tasks.

## G    SAMPLING EXAMPLES

In this section, we present qualitative sampling results to illustrate the performance of our method across different inverse imaging tasks. We provide side-by-side comparisons on both FFHQ and ImageNet datasets under various degradations, including motion blur, Gaussian blur, nonlinear blur, super-resolution, and inpainting. Each figure shows the measurement, reconstructions from DPS and WDPS, and the corresponding ground-truth image. The results highlight the superior visual fidelity of WDPS, especially in challenging scenarios where fine structures and high-frequency details need to be preserved.

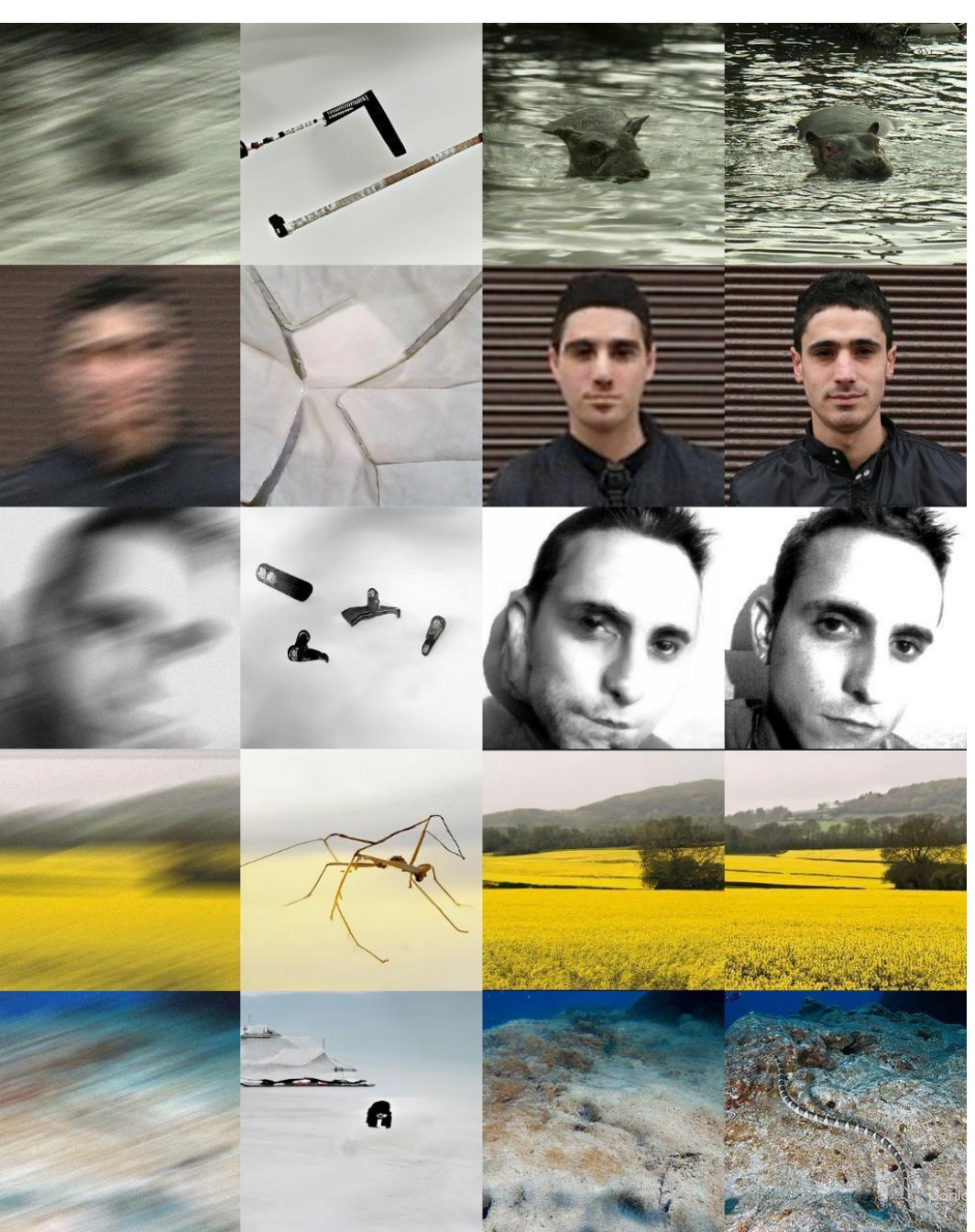

Figure 5: Motion Blur Ablation on ImageNet Dataset: Each row shows a sample, where the images from left to right are **Measurement**, **Non-strength**, **Strength**, and **Ground Truth**.

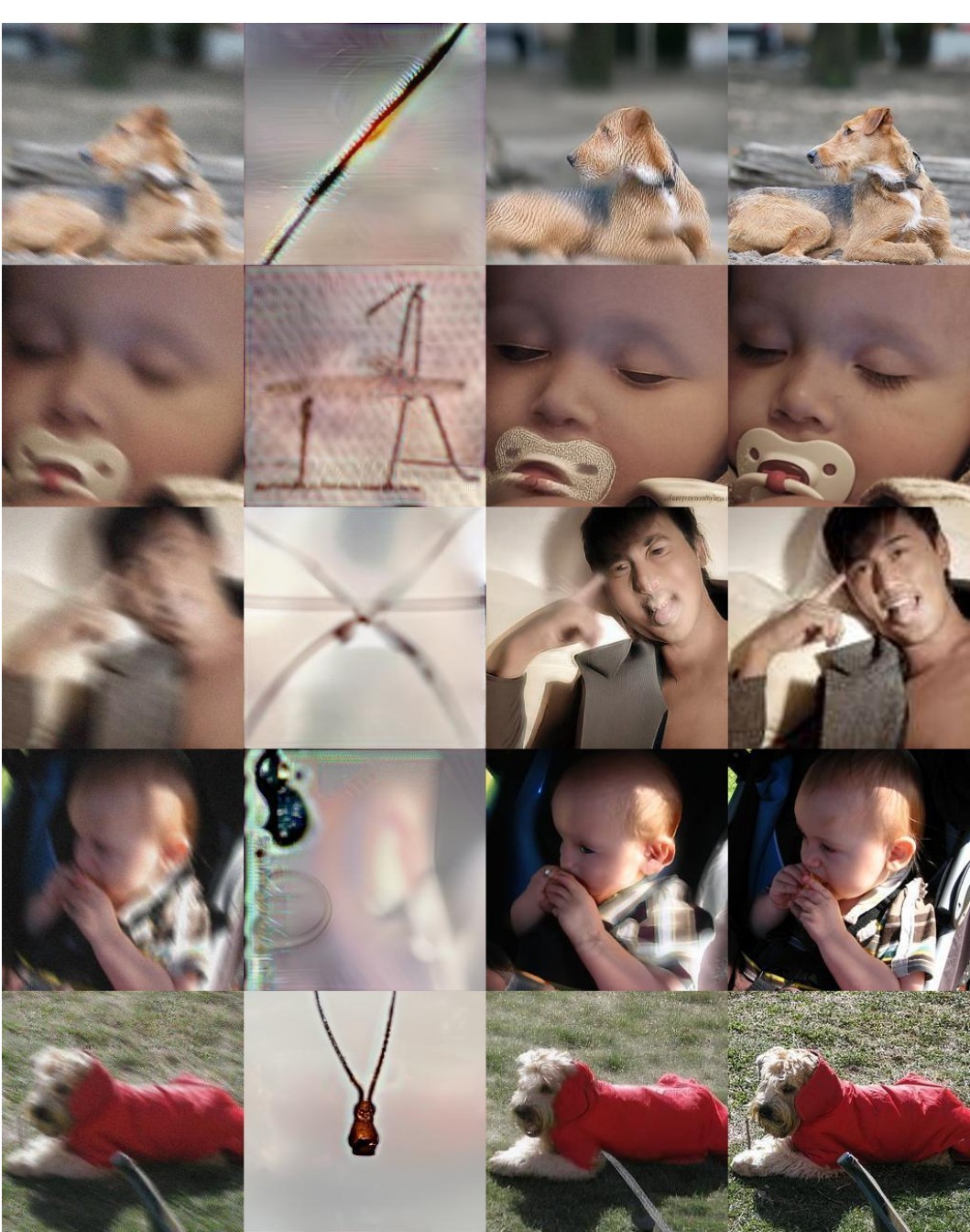

Figure 6: Nonlinear Blur Ablation on ImageNet Dataset: Each row shows a sample. From left to right: **Measurement**, **Strength**, **Non-strength**, **Ground Truth**.

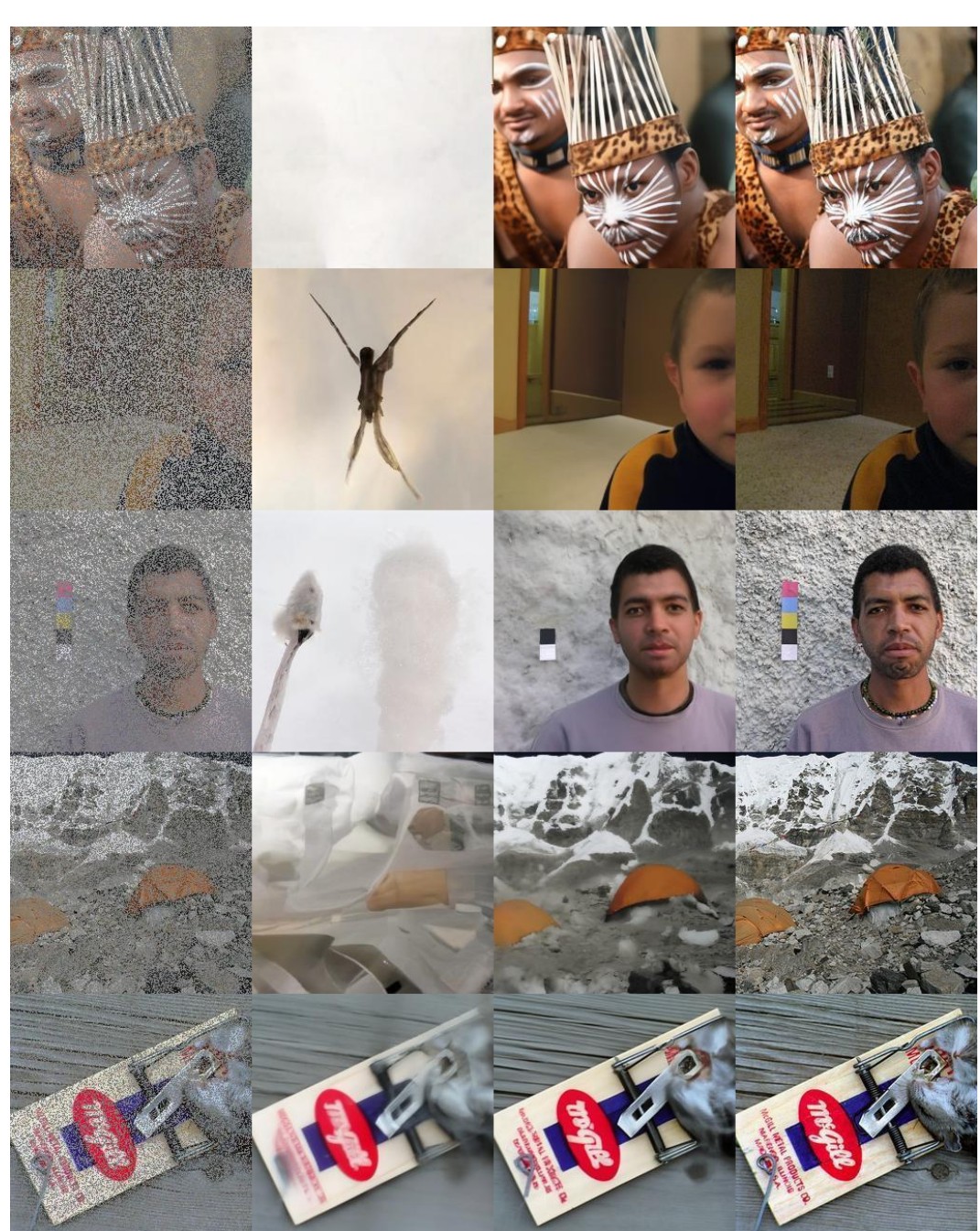

Figure 7: Inpainting Ablation on ImageNet Dataset: Each row shows a sample. From left to right: **Measurement**, **Non-strength**, **Strength**, and **Ground Truth**.

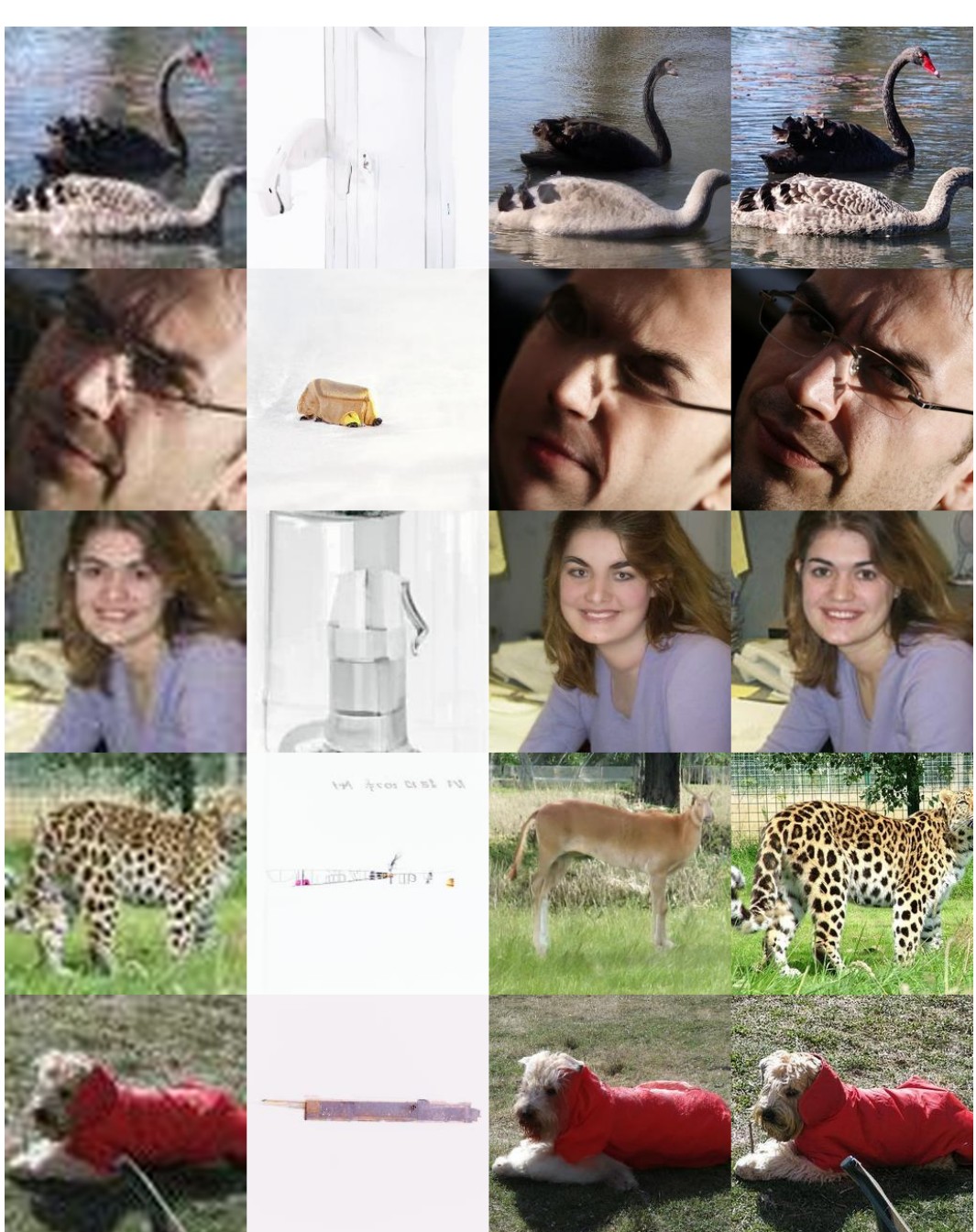

Figure 8: Super-resolution Ablation on ImageNet Dataset: Each row shows a sample. From left to right: **Measurement**, **Non-strength**, **Strength**, and **Ground Truth**.

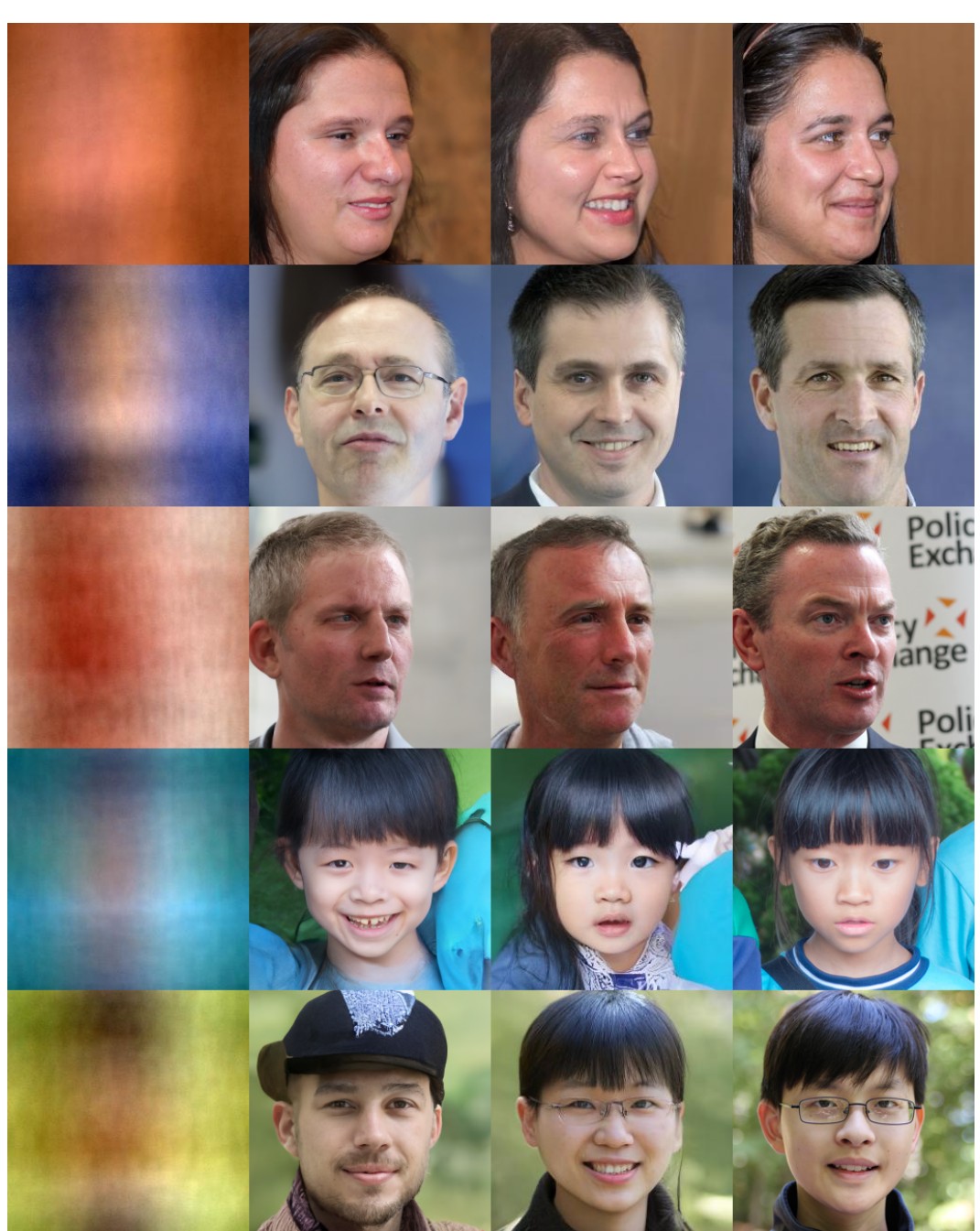

Figure 9: FFHQ DiffuserCam Ablation: Each row shows a sample. From left to right: **Measurement**, **Non-strength**, **Strength**, and **Ground Truth**.

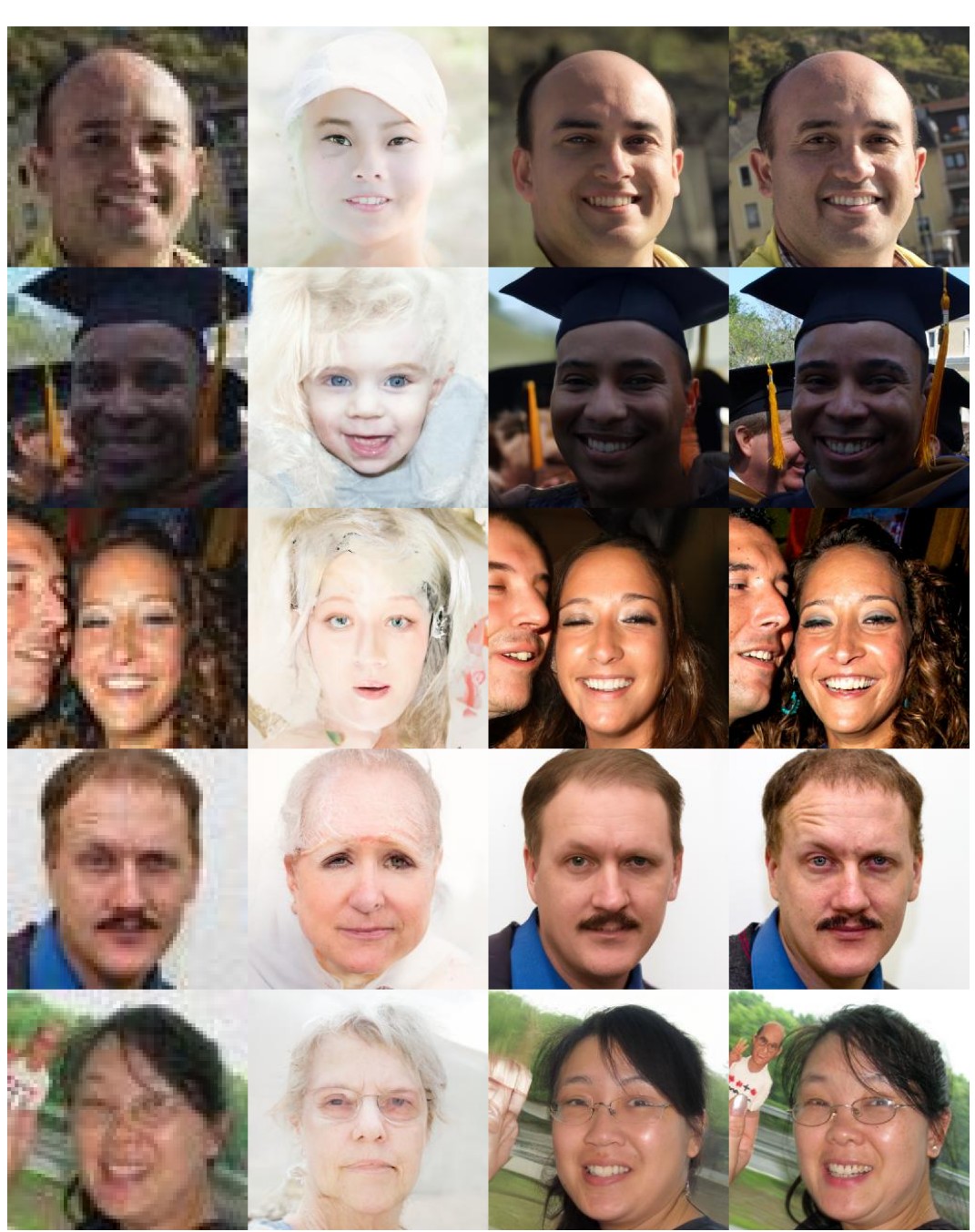

Figure 10: FFHQ Super Resolution Ablation: Each row shows a sample. From left to right: **Measurement**, **Non-strength**, **Strength**, and **Ground Truth**.

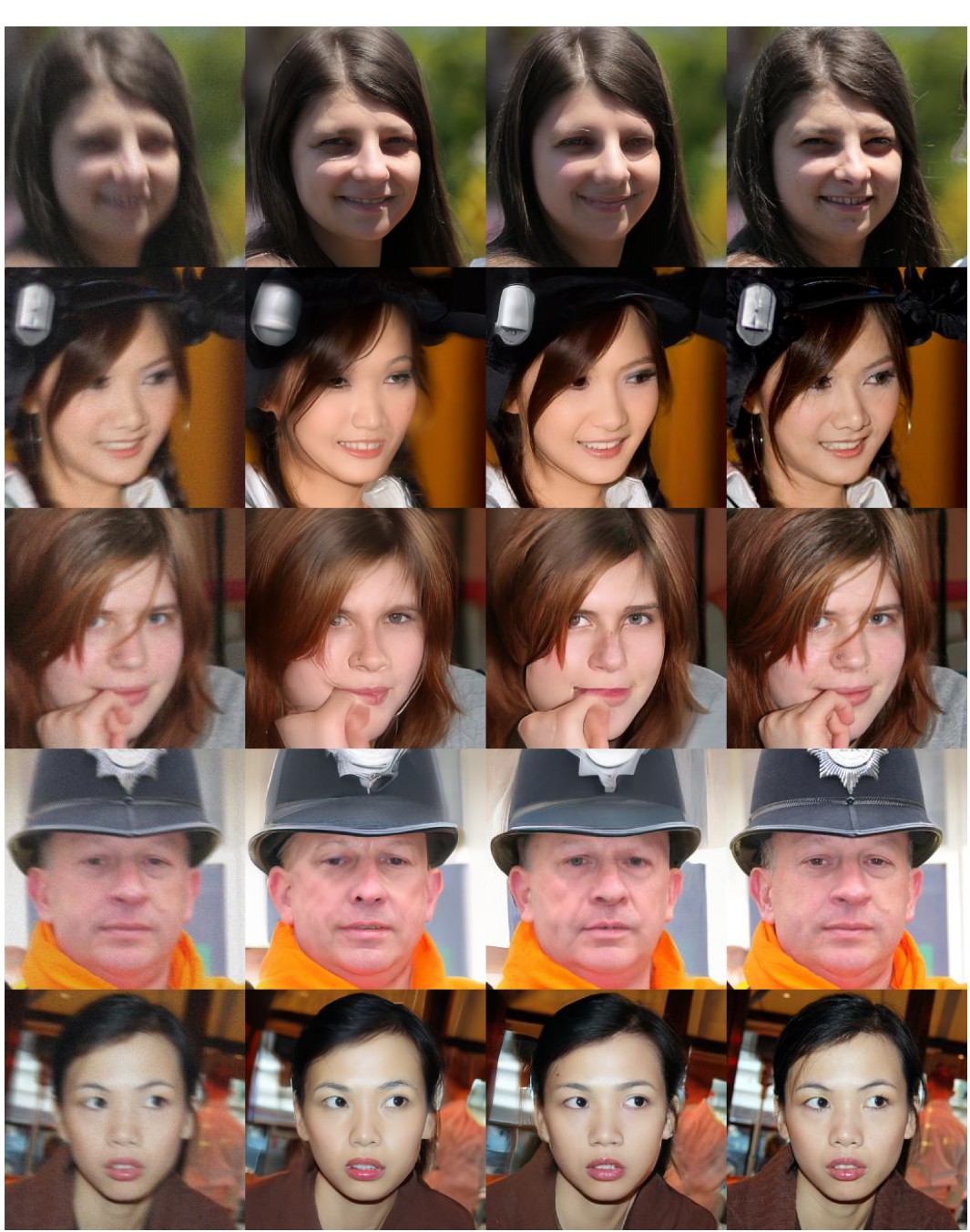

Figure 11: FFHQ Nonlinear Blur Ablation: Each row shows a sample. From left to right: **Measurement**, **Non-strength**, **Strength**, and **Ground Truth**.

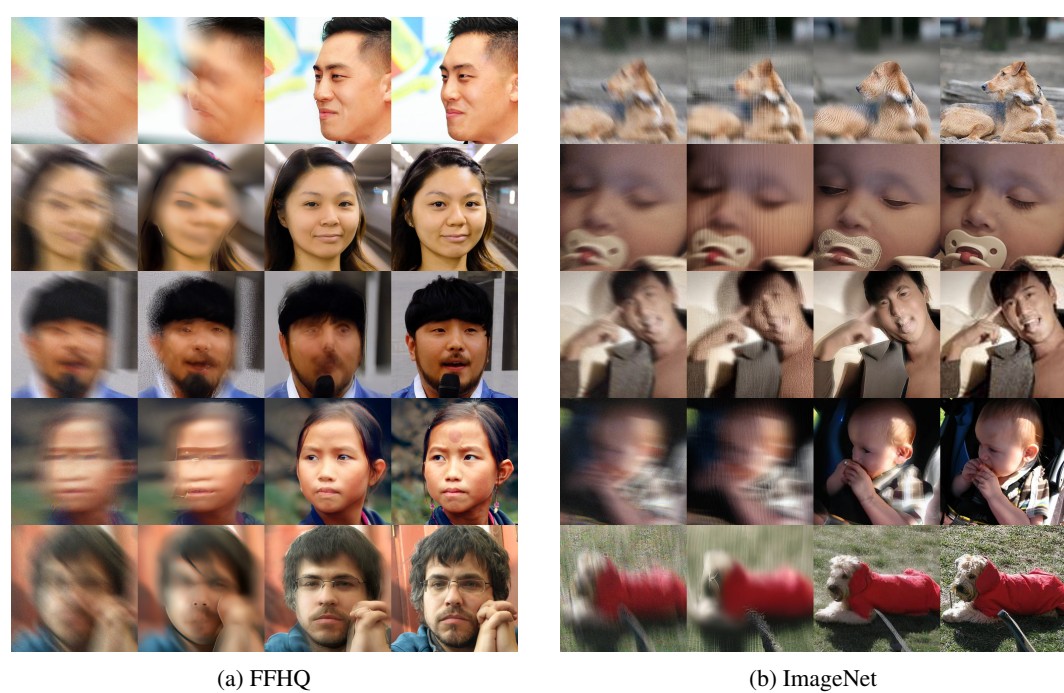

(a) FFHQ                 (b) ImageNet

Figure 12: Nonlinear Blur Ablation on ImageNet Dataset: Each row shows a sample. From left to right: **Measurement**, **DPS**, **WDPS**, and **Ground Truth**.

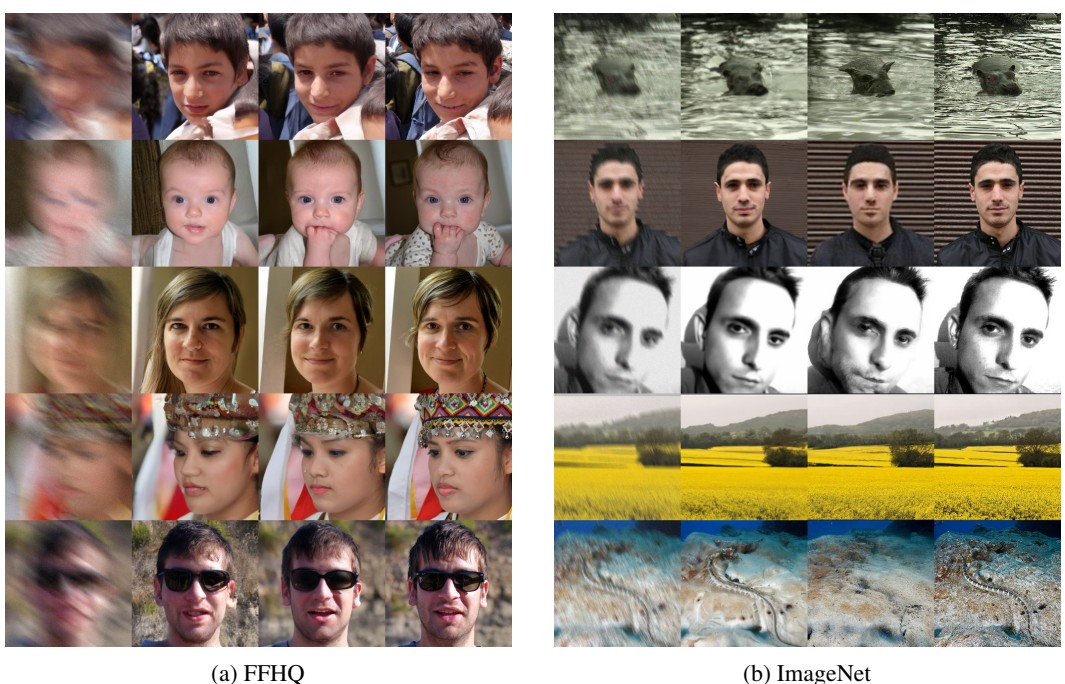

(a) FFHQ                 (b) ImageNet

Figure 13: Motion Blur Ablation: Each row shows a sample. From left to right: **Measurement**, **DPS**, **WDPS**, and **Ground Truth**.

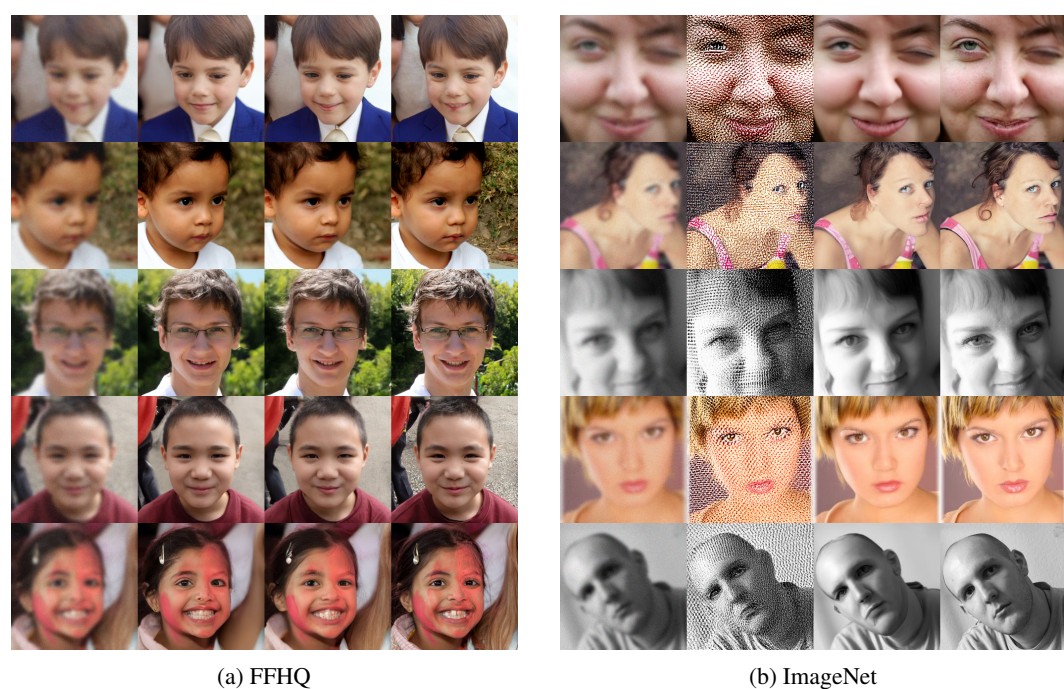

(a) FFHQ                                      (b) ImageNet

Figure 14: Gaussian Blur Ablation: Each row shows a sample. From left to right: **Measurement**, **DPS**, **WDPS**, and **Ground Truth**.

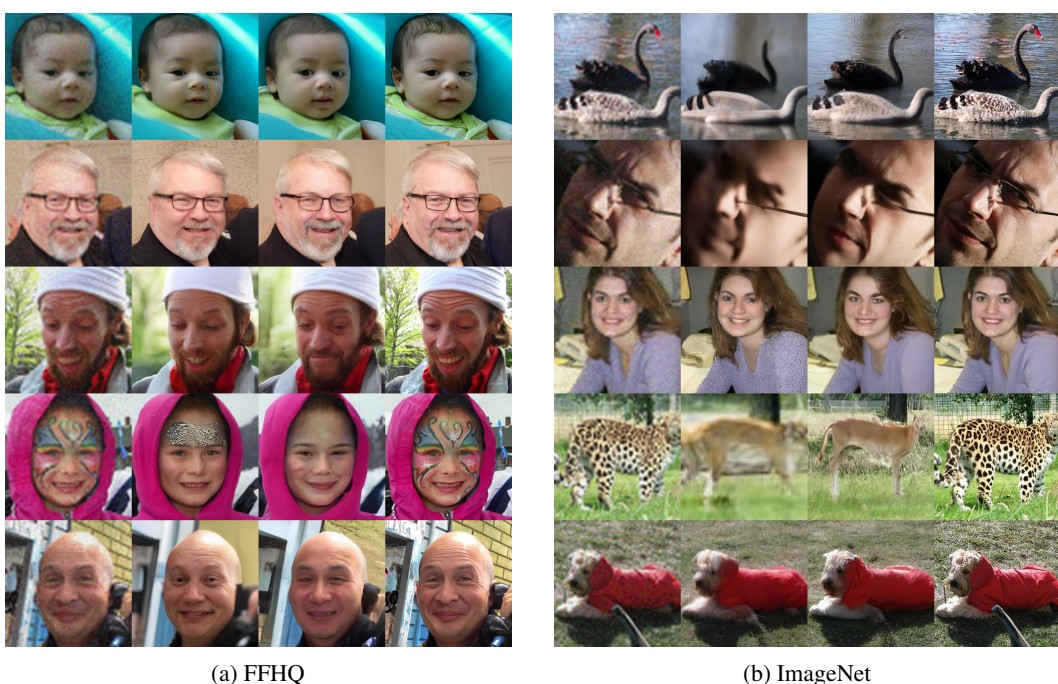

(a) FFHQ                                      (b) ImageNet

Figure 15: Super Resolution Ablation: Each row shows a sample. From left to right: **Measurement**, **DPS**, **WDPS**, and **Ground Truth**.

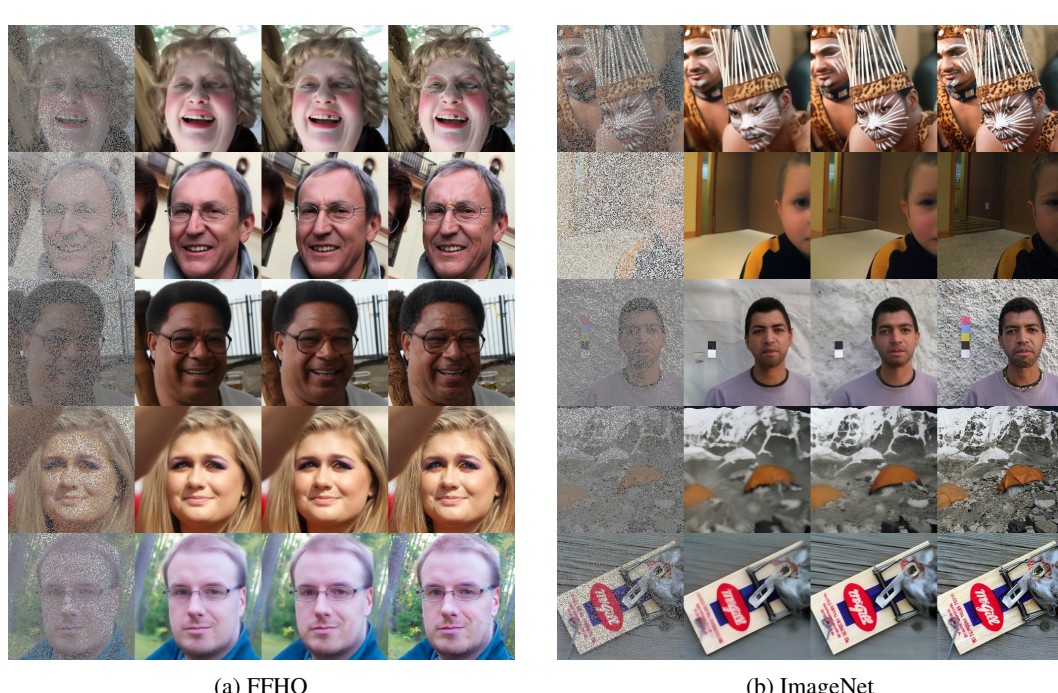

(a) FFHQ                                    (b) ImageNet

Figure 16: Inpainting Ablation: Each row shows a sample. From left to right: **Measurement**, **DPS**, **WDPS**, and **Ground Truth**.

