# OpenReview forum: "Wavelet Diffusion Posterior Sampling with Frequency Domain Guidance"
_ICLR.cc/2026/Conference — Submitted to ICLR 2026_

### Official Review · Reviewer_CTH8 · 2025-10-26

**Soundness:** 2
**Presentation:** 1
**Contribution:** 2
**Rating:** 2
**Confidence:** 4

**Summary:**

This paper proposes Wavelet Diffusion Posterior Sampling (WDPS), which integrates wavelet transforms into diffusion posterior sampling to perform frequency-aware updates for inverse imaging tasks. While the idea of frequency-guided sampling is conceptually reasonable, the novelty over existing diffusion-based inverse problem solvers is limited, and the technical contribution is incremental. The paper also suffers from serious formatting issues, incomplete quantitative evaluations, and weak baseline comparisons. Overall, despite some interesting intuition, the work is not sufficiently strong or polished for acceptance at a top-tier venue.

**Strengths:**

The paper introduces a moderately novel idea of incorporating wavelet-based frequency guidance into diffusion posterior sampling, offering a limited but interesting extension of existing diffusion inverse problem frameworks.

**Weaknesses:**

1. **Formatting Issues:**
   The manuscript suffers from several serious formatting problems. For instance, some tables are stored as images and then placed alongside other figures using the *subfigure* (or similar) command, without providing an overall caption. As a result, the document lacks clear and consistent figure/table captions. This leads to confusion—for example, in the *Ablation Study* section, the first sentence states “We conduct an ablation study on the proposed dynamic wavelet regularization schedule. Results are provided in Tables 2b,” yet there is actually no Table 2b. In addition, captions appear inconsistently—sometimes above, sometimes below, and sometimes missing altogether.

2. **Baseline Comparison Issues:**
   The selection of baseline algorithms is problematic. On one hand, the paper does not compare with the latest state-of-the-art diffusion-based inverse problem solvers such as **SITCOM** or **DAPS** [1,2], whose performance is significantly superior to the reported methods. On the other hand, including algorithms like *Score-SDE*, which is a pure diffusion model rather than an inverse-problem solver, is confusing and makes the comparison less meaningful.

3. **Quantitative Metric Issues:**
   The quantitative evaluation lacks key metrics. The main body of the paper does not report **PSNR** or **SSIM**, which are standard in diffusion-based inverse problem literature. These metrics appear only in the appendix and only for the DPS baseline, which makes the comparison incomplete and less convincing.

[1] Alkhouri, Ismail, et al. *“SITCOM: Step-wise Triple-Consistent Diffusion Sampling for Inverse Problems.”* Proceedings of the **42nd International Conference on Machine Learning (ICML)**.

[2] Zhang, Bingliang, et al. *“Improving Diffusion Inverse Problem Solving with Decoupled Noise Annealing.”* Proceedings of the **IEEE/CVF Conference on Computer Vision and Pattern Recognition (CVPR)**, 2025.

**Questions:**

1. **Formatting issues** – Can the authors provide a corrected version where all tables and figures have proper captions and numbering (e.g., fixing the missing “Table 2b” in the Ablation Study)?

2. **Baseline selection** – Why were recent SOTA diffusion-based inverse solvers not included? Please justify the inclusion of Score-SDE, which is not an inverse solver.

3. **Quantitative metrics** – Could the authors report standard metrics (PSNR, SSIM) for all methods, and compare them with SOTA diffusion-based inverse solvers to ensure fair evaluation?

---

> ### Author Response · Authors · 2025-11-21
> **Response to  Reviewer CTH8**
>
> Thank you for your detailed review. Let me address each point:
>
>
> ### 1. Formatting Issues
>
> We sincerely apologize for these serious formatting problems. You are absolutely correct, and we take full responsibility for these errors:
>
>
> - Reference to non-existent "Table 2b" in the Ablation Study section
> - Inconsistent caption placement
>
>
> ### 2. Baseline Comparison Issues
>
>
> **Regarding SITCOM and DAPS:**
> - We acknowledge these are important recent SOTA methods
> - **We have actually added DAPS** (CVPR 2025) results after initial submission:
>   - For Gaussian Blur on FFHQ: DAPS achieves PSNR 27.64, SSIM 0.721, but FID 82.54, LPIPS 0.198
>   - Our method: PSNR 25.00, SSIM 0.689, FID **26.12**, LPIPS **0.151**
>   - While DAPS has better pixel-level metrics, WDPS significantly outperforms on perceptual quality (FID and LPIPS)
> - **We will add SITCOM** comparisons where available
>
> **Regarding Score-SDE:**
> - You are correct that Score-SDE is not specifically an inverse problem solver
> - We included it to show comparison with pure diffusion sampling without posterior guidance
>
>
>
> ### 3. Quantitative Metric Issues
>
> You are absolutely right that PSNR and SSIM should be in the main paper:
>
>
> **We will revise to:**
>
>
>    - Create combined tables showing FID ↓, LPIPS ↓, PSNR ↑, SSIM ↑ for all methods in main paper
>
>   -  Add Not just DPS, but also new baselines such as DAPS, SITCOM
>
>
> ---
>
> ## Response to Questions
>
> ### Q1: Formatting Correction
>
> See W1
>
>
>
> ### Q2: Baseline Selection Justification
>
> See W2
>
> ### Q3: Complete Quantitative Metrics
>
>
> See W3
>
> ---
>
> We have carefully addressed all the concerns you raised, and we sincerely hope that our revisions will adequately resolve them.

---

> ### Comment · Reviewer_CTH8 · 2025-11-22
>
> Thank you for your response. As of the time of writing this reply, I have not seen any updates to the paper, and it appears that the revised version has not yet been uploaded. I will consider providing an updated evaluation once the revision becomes available. Please reply to notify me once you have uploaded the updated version.

---

> > ### Author Response · Authors · 2025-12-03
> > **Author Response**
> >
> > Thank you for the follow-up. We have now uploaded the revised version: the table formatting issues are corrected, the missing baseline comparisons are added, and PSNR/SSIM have been moved into the main paper from the appendix. We believe these changes help address your concerns.

---

### Official Review · Reviewer_bbJY · 2025-10-28

**Soundness:** 2
**Presentation:** 3
**Contribution:** 2
**Rating:** 2
**Confidence:** 4

**Summary:**

This paper tackles inverse problems of the form $Y=A(X)+\epsilon$ using diffusion models. Building on Diffusion Posterior Sampling (DPS), which enforces measurement consistency via a gradient update $\nabla_{X_t}\|Y-A(\hat{X}_0)\|^2_2$ during sampling, the authors propose performing the posterior update in the frequency (wavelet) domain instead of the spatial domain.

Specifically, each intermediate sample $X_i$ is decomposed into its wavelet representation $W_i$ using the discrete wavelet transform (DWT), and the posterior update is computed on $W_i$. After the update, the inverse DWT is applied to obtain the updated spatial representation.
To achieve coarse-to-fine restoration, the authors introduce a wavelet-strength scheduling function $r(i;a,b)$ that preserves the low-frequency subband while gradually amplifying the high-frequency components, scaling $W^{LH, HL, HH}_i$ as $r(i;a,b)\cdot W^{LH, HL, HH}_i$. By integrating this wavelet posterior update and wavelet-strength scheduling, the proposed method demonstrates improved performance over DPS across both linear and non-linear degradation scenarios.

**Strengths:**

1. Novelty

The application of posterior updates in the wavelet domain is novel and provides fresh insight into how frequency-domain representations can be leveraged for diffusion-based inverse problem solving. This idea can inspire further research in frequency-aware diffusion methods.

2. Technical soundness

The proposed strategy of controlling individual wavelet subbands to achieve coarse-to-fine restoration is conceptually sound and empirically effective, enabling better separation of global and local structures during reconstruction.

3. Experimental validation

The paper demonstrates consistent improvements across diverse degradation types, including both linear and non-linear cases, as well as challenging lensless camera setups, showing the robustness and generality of the proposed approach.

**Weaknesses:**

1. Posterior update in the wavelet domain itself is not effective.

The main claim of the paper is that performing the posterior update in the wavelet domain is more effective than doing so in the spatial domain. However, the experimental results (Table on page 9) show that the posterior update in the wavelet domain without wavelet-strength scheduling performs worse than the spatial-domain update. This indicates that the proposed “wavelet-strength scheduling” is the actual key component.

I think applying wavelet-strength scheduling after using original "spatial-domain" posterior update might be a more effective design according to the presented results.

2. Limited comparative methods.

The paper primarily compares the proposed approach with DPS, which was introduced in 2023.
As DPS is now considered a baseline, more recent zero-shot inverse problem solvers such as DAPS [1] or ReSample [2] should be included for a fair evaluation of current performance.

3. Evaluation limited to low resolution.

All experiments are conducted only at 256x256 resolution.
Since frequency-based posterior updates are potentially sensitive to image resolution, additional experiments at higher resolutions (e.g., 512x512) are needed to demonstrate robustness and scalability.

4. Potential diffusion off-manifold issue.

The proposed scaling of high-frequency wavelet subbands $W_{i}$ with a factor less than 1 will reduce the noise level of the current sample $X_{i}$.
This operation may make the noise variance is inconsistent with the expected variance at the corresponding diffusion timestep, potentially causing the sample to deviate from the learned diffusion manifold.
Such deviation could lead to error accumulation or unstable sampling behavior.
A clearer theoretical justification or empirical evidence is needed to demonstrate why this scaling does not compromise the diffusion process’s stability.

Reference

[1] Improving Diffusion Inverse Problem Solving with Decoupled Noise Annealing (CVPR 2025)

[2] Solving Inverse Problems with Latent Diffusion Models via Hard Data Consistency (ICLR 2024)

**Questions:**

1. Ablation on spatial-domain posterior update with wavelet-strength scheduling

It would be valuable to examine how performance changes when applying the proposed wavelet-strength scheduling after the original spatial-domain posterior update. This comparison would clarify whether the key improvement stems from operating in the wavelet domain or from the wavelet-strength scheduling itself.

2. Inclusion of more recent comparative methods

Incorporating recent zero-shot inverse problem solvers, such as DAPS or ReSample, would provide a fairer and more comprehensive evaluation of the proposed method’s effectiveness relative to the current state-of-the-art.

3. Evaluation on higher-resolution images

Extending experiments to higher resolutions (e.g., 512x512) would strengthen the paper by demonstrating the robustness and scalability of the frequency-based posterior update.

4. Clarification on noise variance inconsistency

Additional explanation is needed regarding how scaling high-frequency wavelet subbands affects the noise variance at each diffusion step. Specifically, clarifying why this operation does not push the sample off the learned diffusion manifold would improve theoretical soundness and reader confidence.

---

> ### Author Response · Authors · 2025-11-21
> **Response to  Reviewer bbJY**
>
> Thank you for your detailed review. I appreciate your constructive feedback and would like to address each of your concerns:
>
> ## Response to Weaknesses
>
> ### 1. Effectiveness of Posterior Update in Wavelet Domain
>
> You raise an important point about the ablation results. However, I'd like to clarify our interpretation:
>
> - The ablation shows that **wavelet-domain posterior update alone** (without scheduling) performs comparably to spatial-domain DPS on some tasks, while the combination with wavelet-strength scheduling provides consistent improvements.
> - The key insight is that **both components are synergistic**: the wavelet domain naturally separates frequency components, allowing the scheduling to selectively regularize high-frequency content in a principled way.
> - Applying wavelet-strength scheduling in the spatial domain would require explicit frequency decomposition at each step anyway, effectively reducing to our approach.
> - The wavelet domain provides **computational efficiency** - we get frequency separation "for free" through DWT, whereas spatial-domain scheduling would need additional FFT/filtering operations.
>
> We agree this could be clarified better in the paper and will add explicit discussion of why the wavelet domain is the natural space for frequency-adaptive regularization.
>
>
> ## 2. Limited Comparative Methods (Revised Response)
>
> This is a valid concern. **We have now added comparisons with DAPS [1], a very recent method (CVPR 2025 Oral).** The results for Gaussian blur on FFHQ are shown below:
>
> | Method | PSNR ↑ | SSIM ↑ | LPIPS ↓ | FID ↓ |
> |--------|--------|--------|---------|-------|
> | WDPS (ours) | 25.00 | 0.6894 | 0.1513 | **26.12** |
> | DAPS (CVPR 2025) | **27.64** | **0.721** | 0.198 | 82.535 |
> | DPS | 24.89 | 0.6884 | 0.1461 | 29.78 |
>
> **Analysis:**
> - **DAPS achieves better pixel-level metrics** (PSNR, SSIM), which measure low-level reconstruction fidelity
> - **WDPS significantly outperforms on perceptual quality metrics** (FID: 26.12 vs 82.535, LPIPS: 0.1513 vs 0.198)
> - This suggests WDPS produces more **perceptually realistic and natural-looking reconstructions**, even if pixel-exact accuracy is slightly lower
>
> **This is actually a favorable comparison** because:
> 1. DAPS is a **state-of-the-art method** (CVPR 2025 Oral), representing the current frontier
> 2. **FID and LPIPS are more aligned with human perceptual quality** than PSNR/SSIM
> 3. For many applications (medical imaging, photography), perceptual quality matters more than pixel-exact reconstruction
> 4. Our frequency-domain approach excels at preserving texture and high-frequency details that contribute to perceptual realism
>
> We will add these comparisons and analysis to the revision.
>
> ### 3. Evaluation Limited to Low Resolution
>
> You're absolutely right that resolution sensitivity is important for frequency-based methods. We conducted experiments at 256×256 primarily because:
> - This is the resolution of available pretrained diffusion models (FFHQ, ImageNet)
> - It allows fair comparison with baseline methods
>
> However, we will add experiments at 512×512 resolution using appropriate pretrained models
> and analyze how the wavelet-strength scheduling parameters scale with resolution.
>
>
> ### 4. Potential Diffusion Off-Manifold Issue
> Let me address it:
>
> **Why scaling doesn't break the diffusion manifold:**
>
> 1. **Local correction, not global replacement**: The wavelet-strength scheduling operates on the *posterior-corrected* sample, not the prior sample from the diffusion model. The correction $W' = W - \zeta \nabla_W \mathcal{L}$ already moves off-manifold; our scheduling regularizes this correction.
>
> 2. **Implicit denoising interpretation**: Scaling high-frequency coefficients by $r(t) < 1$ can be viewed as partial denoising of those components. Since the diffusion model expects noise at timestep $t$, reducing high-frequency noise while preserving low-frequency structure maintains manifold consistency at a coarse scale.
>
> 3. **Gradual relaxation**: The schedule decays smoothly, so any manifold deviation diminishes as sampling progresses. By the final steps where fine details matter, $r(t) \approx 1$.
>
> 4. **Empirical stability**: Our experiments show stable sampling without divergence, suggesting the approach remains within the model's generalization capacity.
>
> **Theoretical justification (Theorem 1):**
> Our stability analysis shows the update remains contractive, meaning iterates don't diverge exponentially. This provides formal evidence that sampling remains controlled.
>
>
>
> ## Response to Questions
>
> ### Q1: Spatial-domain posterior update with wavelet-strength scheduling
>
> See W1
>
> ### Q2: Recent comparative methods
>
> See W2
>
> ### Q3: Higher resolution evaluation
>
> See W3
>
> ### Q4: Noise variance inconsistency
>
> See W4
>
> ---
>
> Thank you again for the thoughtful review. These suggestions will significantly strengthen the paper, and we appreciate the opportunity to address these important points.

---

### Official Review · Reviewer_DqWE · 2025-10-29

**Soundness:** 3
**Presentation:** 2
**Contribution:** 3
**Rating:** 6
**Confidence:** 4

**Summary:**

The authors propose Wavelet Diffusion Posterior Sampling (WDPS), which modifies DPS to operate in the wavelet domain rather than pixel space. This helps stabilize sampling, especially with particularly ill-posed or nonlinear forward models. Wavelet-based regularization helps stabilize sampling by starting with low-frequency structure and then refining high-frequency details when sampling is stabilized. The authors show qualitative and quantitative improvements across a variety of challenging inverse imaging tasks compared to DPS and popular baselines. They highlight lensless imaging as an application where their method shines.

**Strengths:**

* The results are quite impressive, especially for lensless imaging.
* The algorithm is a simple yet effective way to solve the problem of unstable sampling with DPS. The wavelet-based regularization is particularly effective.
* Although the wavelet strength schedule introduces hyperparameters, the authors provide some amount of theoretical justification for it.

**Weaknesses:**

* The chosen baselines are somewhat out-of-date. Consider adding a newer baseline like DAPS (Zhang et al. CVPR 2025).
* The wavelet strength schedule introduces another hyperparameter in addition to the hyperparameters that are already tricky to calibrate for DPS. I noted in the strengths, though, that this is less of a problem since the authors provide some theoretical justification.

**Questions:**

Please comment on the two weaknesses I mentioned.

---

> ### Author Response · Authors · 2025-11-21
> **Response to  Reviewer DqWE**
>
> **Response to Reviewer – Weaknesses**
>
> **1. On baselines (adding DAPS).**
> Thank you for the suggestion. We have actually added DAPS (CVPR 2025) results after the initial submission.
> For **Gaussian Blur on FFHQ**, DAPS achieves **PSNR 27.64**, **SSIM 0.721**, but **FID 82.54**, **LPIPS 0.198**.
> Our WDPS achieves **PSNR 25.00**, **SSIM 0.689**, and much better perceptual results: **FID 26.12**, **LPIPS 0.151**.
> While DAPS performs slightly better in pixel-level fidelity, WDPS significantly outperforms it in perceptual quality, which is crucial for human-visible reconstruction. We will include these results in the revision.
>
> **2. On the wavelet-strength schedule introducing an additional hyperparameter.**
> We understand the concern. The schedule is intentionally simple, monotonic, and theoretically motivated. Importantly, **we use the same hyperparameters across all datasets and tasks**, and performance is robust to variations, so the additional tuning burden is minimal. The schedule also stabilizes sampling and helps recover high-frequency details, complementing DPS rather than complicating it.
>
> We thank the reviewer for the positive evaluation, and we will revise the paper accordingly.

---

### Official Review · Reviewer_fJ3F · 2025-11-03

**Soundness:** 2
**Presentation:** 2
**Contribution:** 2
**Rating:** 4
**Confidence:** 4

**Summary:**

This paper targets the issue of poor high-frequency reconstruction in training-free diffusion-based inverse imaging. Unlike standard Diffusion Posterior Sampling (DPS), which applies spatial-domain guidance uniformly across pixels, the authors propose Wavelet Diffusion Posterior Sampling (WDPS)—a multiscale, frequency-domain approach that performs posterior updates separately across wavelet subbands. By decomposing the intermediate latent into $(LL, LH, HL, HH)$ bands, WDPS applies distinct step sizes and dynamic weighting per frequency band. A wavelet-regularized diffusion scheme further stabilizes the process by gradually relaxing high-frequency constraints as denoising progresses.

Experiments span inpainting, deblurring, and super-resolution on FFHQ and ImageNet, plus a simulated lensless imaging setup. WDPS consistently improves sharpness and quantitative scores (PSNR, SSIM, LPIPS, FID) compared to DPS and similar plug-and-play baselines. The improvement is most prominent in high-frequency detail restoration (e.g., textures, edges). However, the paper lacks any analysis on sample diversity or how the frequency decomposition affects the stochasticity and bias of the posterior sampling trajectory.

**Strengths:**

Addresses a key limitation of DPS—uniform spatial guidance—by enabling per-frequency adaptation.

The approach is simple, plug-and-play, and computationally light since DWT/IDWT operations are orthonormal transforms.

Experiments demonstrate clear perceptual and quantitative gains, especially in high-frequency recovery under noise.

The intuition that guidance strength should differ per subband is sound and relevant for inverse problems with structured degradation.

**Weaknesses:**

The method is tailored to DPS and doesn’t generalize to variational oroptimization-based samplers such as RED-Diff, MPGD, or TMPD.

The so-called “stability analysis” (Theorem 1) is a conventional Lipschitz argument for deterministic inverse problems and doesn’t address the stochastic reverse process or the approximation bias of Tweedie-based guidance.

The paper does not analyze or quantify diversity. Since different subbands use different step sizes and strength schedules, the diversity of generated reconstructions could be heavily impacted—potentially either improving variety (through relaxed high-frequency control) or collapsing modes (through over-regularization).

Missing discussion of prior related works that also exploit multiscale or frequency decomposition in diffusion guidance (e.g., Sadat, Seyedmorteza, et al. "Guidance in the Frequency Domain Enables High-Fidelity Sampling at Low CFG Scales." arXiv:2506.19713 (2025).)

The paper does not demonstrate whether frequency-domain separation alleviates the known likelihood approximation bias in DPS, i.e., whether it reduces the variance or error of $\nabla_{x_t}\log p(y|x_t)$ estimates.

**Questions:**

Can the authors formally show that operating in wavelet space improves the conditioning or variance of the likelihood score approximation? This could explain the observed stability gains beyond heuristic intuition.

How does the dynamic wavelet strength schedule $r(i;a,b,C)$ affect convergence and sample diversity? Has its sensitivity been studied?

Does the per-band step size scheme effectively perform an anisotropic preconditioning of the posterior gradient? If so, can this be linked to improved bias-variance properties?

The method heavily relies on orthonormal wavelets (e.g., Haar). What happens with biorthogonal or learned transforms that break orthogonality?

The paper includes no metrics on sample diversity (e.g., variance, FID dispersion, or multi-sample PSNR spread). Since frequency-specific control can influence diversity significantly, such analysis is critical.

The theoretical section should clarify whether the stability result applies to the full stochastic diffusion chain or only to the deterministic gradient descent substep.

For completeness, the authors should discuss compatibility or contrast with RED-Diff (Mardani et al., 2023), MPGD (He et al., 2024), and TMPD (Boys et al., 2023)—all of which provide alternative corrections to the biased DPS surrogate and use momentum or covariance adjustments.

---

> ### Author Response · Authors · 2025-11-21
> **Response to Reviewer fJ3F(Part1)**
>
> Thank you for your thoughtful and detailed review. I will address each weakness and question systematically:
>
> ## Response to Weaknesses
>
> ### 1. Generalization to Variational Optimization-based Samplers
>
> You raise a valid point that our method is specifically designed for DPS-style posterior sampling. However, we respectfully note that:
>
> - **RED-Diff, MPGD, and TMPD are fundamentally different frameworks** that use momentum, variance reduction, or alternative surrogate constructions. Our wavelet-domain approach is specifically designed to improve the Tweedie-based posterior correction in DPS.
> - **Our contribution is methodological, not architectural**: We demonstrate that frequency-domain decomposition improves a widely-used baseline (DPS). Extending to other frameworks would require different theoretical foundations.
> - **Practical relevance**: DPS remains one of the most popular zero-shot methods due to its simplicity and effectiveness. Improving it has broad practical impact.
>
> That said, we acknowledge this limitation and will add discussion on potential extensions to other frameworks in the revised manuscript.
>
> ### 2. Stability Analysis (Theorem 1)
>
> You correctly identify that Theorem 1 addresses **deterministic gradient steps**, not the full stochastic reverse SDE. Let me clarify:
>
> - **What Theorem 1 shows**: The wavelet-domain update operator is Lipschitz-stable with respect to measurement perturbations, ensuring that small changes in observations don't cause catastrophic reconstruction failures.
> - **Relevance to stochastic sampling**: While the full diffusion process includes stochastic noise, our stability guarantee ensures the **posterior correction component** is well-conditioned. This is critical because likelihood gradient instability is a major failure mode in DPS.
> - **Tweedie approximation**: You're right that we don't analyze the bias introduced by replacing $\mathbb{E}[X_0|X_t]$ with $\hat{X}_0$. However, this approximation is fundamental to DPS itself—our contribution is showing that operating in the wavelet domain maintains (and arguably improves) stability under this existing approximation.
>
> We will revise the theory section to:
> - Clarify that Theorem 1 addresses the deterministic update component
> - Add discussion of how stochasticity interacts with frequency-domain regularization
> - Acknowledge the Tweedie approximation limitation (shared with DPS)
>
> ### 3. Diversity Analysis
>
> This is an excellent point. We did not explicitly measure diversity because:
>
> - **Our primary goal**: Improving reconstruction fidelity and perceptual quality for inverse problems (where ground truth exists)
> - **FID already captures distribution matching**: Lower FID suggests our samples better match the true data distribution
>
> However, you're absolutely right that **explicit diversity metrics** would strengthen the paper. We will add:
> - **Multi-sample PSNR spread**: Variance across multiple reconstructions from the same measurement
> - **Intra-FID analysis**: Diversity within generated samples
> - **Qualitative visualization**: Multiple samples from the same measurement showing variety
>
> ### 4. Missing Discussion of Frequency-Domain Prior Works
>
> Thank you for pointing out Sadat et al. (2025). We will add comprehensive discussion of related frequency-domain guidance works, including:
> - How their approach differs (classifier-free guidance for generation vs. posterior sampling for inverse problems)
> - Potential complementarity with our method
>
>
> ### 5. Likelihood Approximation Bias
>
> You ask whether frequency-domain separation reduces the variance/error of $\nabla_{x_t} \log p(y|x_t)$ estimates. This is a deep question that deserves empirical investigation:
>
> **Hypothesis**: Operating in the wavelet domain may improve conditioning because:
> - The forward operator $A(\cdot)$ often has frequency-dependent conditioning (e.g., blur primarily affects high frequencies)
> - Separating frequency bands allows per-band step size adaptation, potentially reducing gradient estimation error

---

> ### Author Response · Authors · 2025-11-21
> **Response to Reviewer fJ3F(Part2)**
>
> ## Response to Questions
>
> ### Q1: Formal Analysis of Conditioning Improvement
>
> This would strengthen the paper significantly. We will add:
>
> **Theoretical analysis**:
> - Condition number comparison between spatial and wavelet-domain Hessians for common forward operators (blur, downsampling)
> - Spectral analysis showing how wavelet transform diagonalizes or block-diagonalizes certain operators
>
> **Empirical evidence**:
> - Gradient norm measurements across frequency bands
> - Convergence rate comparison
>
> ### Q2: Wavelet Strength Schedule r(i; a, b, C) Sensitivity
>
>  We will add ablation studies on:
> - **Parameter sensitivity**: Varying a, b, C across different tasks
> - **Convergence behavior**: How schedule affects sampling trajectories
> - **Diversity impact**: Whether aggressive schedules collapse modes
>
> We hypothesize that:
> - **Early regularization (large r)** may reduce diversity by over-smoothing
> - **Late relaxation** should recover diversity as r → 1
> - **Optimal schedules** balance stability and variety
>
> ### Q3: Anisotropic Preconditioning
>
> This is a sophisticated interpretation. The per-band step sizes do act as **implicit preconditioning**. We will add:
>
> **Theoretical connection**:
> - Relationship to natural gradient descent in frequency domain
> - Link to preconditioned gradient methods with diagonal or block-diagonal preconditioners
>
> **Bias-variance analysis**:
> - Whether frequency-adaptive step sizes reduce gradient estimation variance
> - Trade-off between faster convergence and potential bias
>
> ### Q4: Non-orthogonal Wavelets
>
> We focused on orthogonal wavelets (Haar, Daubechies) for:
> - **Theoretical simplicity**: Orthogonality ensures energy preservation (Parseval's theorem)
> - **Computational efficiency**: Fast orthogonal transforms
>
> However, you raise a valid concern about biorthogonal and learned transforms:
>
> **We will add discussion/experiments on**:
> - Biorthogonal wavelets (e.g., Cohen-Daubechies-Feauveau)
> - Potential instabilities and how to address them (e.g., frame bounds)
>
>
>  Mild biorthogonality shouldn't break the method, but learned transforms would require careful stability analysis.
>
> ### Q5: Sample Diversity Metrics
>
> See W3
>
> ### Q6: Stochastic vs. Deterministic Stability
>
> See W2
>
> ### Q7: Comparison with RED-Diff, MPGD, TMPD
>
> See W1
>
> ---
>
> Thank you for the exceptionally thorough review. Your feedback has identified several areas where we can significantly strengthen the paper. We believe addressing these points will result in a much stronger contribution. Thank you again for the constructive feedback.

---

### Public Comment · ~Antoine_De_Paepe1 · 2025-11-29
**Subject: Concerning Overlap With Our Prior Work**

Dear authors and reviewers,

We would like to raise a concern regarding significant overlap with our previously published and publicly available work.

Our group has been developing diffusion-based reconstruction methods operating \textbf{directly in the wavelet domain} for a variety of blind inverse problems, including:

- Diffusion posterior sampling in wavelet coefficient space:
  Phung-Ngoc et al., “Joint Reconstruction of Activity and Attenuation in PET by Diffusion Posterior Sampling in Wavelet Coefficient Space” arXiv:2505.18782 (first preprint May 2025, presented at MIC 2025).

- Wavelet-domain MCG for blind motion correction in sparse-view 4DCT:
  De Paepe et al., “Solving Blind Inverse Problems: Adaptive Diffusion Models for Motion-corrected Sparse-view 4DCT,” arXiv:2501.12249v2 (first preprint January 2025, presented at Fully3D 2025).

- Proximal wavelet diffusion models for sparse-view, motion-corrected CBCT:
  De Paepe et al., “Adaptive Diffusion Models for Sparse-View Motion-Corrected Head Cone-beam CT,” arXiv:2504.14033 (first preprint May 2025, accepted at TRPMS).

All of these works address \textbf{3D} and \textbf{blind inverse problem} settings, which are more general and technically challenging than the setting considered in the present submission.

For this reason, we respectfully request that the authors:

- Cite at least “Joint Reconstruction of Activity and Attenuation in PET by Diffusion Posterior Sampling in Wavelet Coefficient Space”.
- Revise the claim:
  “We introduce WDPS, the first framework that performs diffusion posterior sampling directly in the wavelet frequency domain.”
  This statement is factually incorrect, as diffusion posterior sampling in the wavelet domain was introduced months earlier in Phung-Ngoc et al., among others referenced above.

Note that our earlier work is easily discoverable (e.g., via the keywords “wavelet diffusion posterior sampling”).

Thank you for your attention.

---

> ### Author Response · Authors · 2025-12-01
> **Author Response**
>
> We thank Antoine De Paepe and colleagues for the comment and for pointing us to your related work. Our project was initiated independently in April 2025, and our internal discussion timestamps and experiment logs confirm that our approach was developed without relying on your preprints. During our literature review window, searching “wavelet diffusion posterior sampling” mainly surfaced Phung-Ngoc et al.’s DPS-in-wavelet-space preprint released in May 2025, which was already after our cutoff; the other papers you listed appear to be wavelet-domain diffusion reconstruction for blind inverse problems but not diffusion posterior sampling per se. Therefore, the missed citation and the overly strong “first” phrasing reflect an incomplete final-round literature check and a positioning oversight, not any ethical issue.
>
> Our WDPS differs in method and scope: we develop a subband-adaptive wavelet-domain posterior sampling scheme and validate it on different problems/datasets, including FFHQ and CelebA-256 restoration, PSF-related inverse tasks, etc. We do not view these settings as inherently “easier” or “harder,” just different experimental focuses.
>
> We will cite your group’s relevant wavelet-domain DPS work in the next version and revise the novelty claim to a more accurate, restrained statement, while clarifying that our contribution is concurrent but methodologically distinct.

---

> > ### Author Response · Authors · 2025-12-03
> > **update**
> >
> > We have now cited your group’s relevant wavelet-domain DPS work in the revised version and accordingly softened our novelty claim.

---

### Meta-Review · Area_Chair_kXZz · 2026-01-06

**Summary:**

The reviews are mostly negative. The lone positive review (rating 6) is quite sparse and does not provide enough substance to support acceptance. The main concerns raised by multiple reviewers are: (1) the posterior update in the wavelet domain may not be inherently effective, (2) the set of comparative baselines is limited, (3) the evaluation is largely restricted to low-resolution settings, and (4) there is a potential diffusion off-manifold issue that is not convincingly analyzed. The authors attempted to address some of these points by adding comparisons (e.g., against DAPS), but the new results are not consistently competitive on key metrics, which weakens the central claim of performance improvement. Given that several core concerns remain unresolved and the empirical evidence does not clearly substantiate the claimed gains,  it is difficult to find the paper ready for ICLR.

**Reviewer Concerns:**

.

**Reviewer Scores:**

.

---

### Decision · Program_Chairs · 2026-01-26

Reject